# Interactions between Dietary Antioxidants, Dietary Fiber and the Gut Microbiome: Their Putative Role in Inflammation and Cancer

**DOI:** 10.3390/ijms25158250

**Published:** 2024-07-28

**Authors:** Camelia Munteanu, Betty Schwartz

**Affiliations:** 1Department of Plant Culture, Faculty of Agriculture, University of Agricultural Sciences and Veterinary Medicine, 400372 Cluj-Napoca, Romania; 2The Institute of Biochemistry, Food Science and Nutrition, The School of Nutritional Sciences, Robert H. Smith Faculty of Agriculture, Food and Environment, The Hebrew University of Jerusalem, Rehovot 7610001, Israel

**Keywords:** antioxidants, dietary fiber, gut microbiome, inflammation, cancer, flavonoids

## Abstract

The intricate relationship between the gastrointestinal (GI) microbiome and the progression of chronic non-communicable diseases underscores the significance of developing strategies to modulate the GI microbiota for promoting human health. The administration of probiotics and prebiotics represents a good strategy that enhances the population of beneficial bacteria in the intestinal lumen post-consumption, which has a positive impact on human health. In addition, dietary fibers serve as a significant energy source for bacteria inhabiting the cecum and colon. Research articles and reviews sourced from various global databases were systematically analyzed using specific phrases and keywords to investigate these relationships. There is a clear association between dietary fiber intake and improved colon function, gut motility, and reduced colorectal cancer (CRC) risk. Moreover, the state of health is reflected in the reciprocal and bidirectional relationships among food, dietary antioxidants, inflammation, and body composition. They are known for their antioxidant properties and their ability to inhibit angiogenesis, metastasis, and cell proliferation. Additionally, they promote cell survival, modulate immune and inflammatory responses, and inactivate pro-carcinogens. These actions collectively contribute to their role in cancer prevention. In different investigations, antioxidant supplements containing vitamins have been shown to lower the risk of specific cancer types. In contrast, some evidence suggests that taking antioxidant supplements can increase the risk of developing cancer. Ultimately, collaborative efforts among immunologists, clinicians, nutritionists, and dietitians are imperative for designing well-structured nutritional trials to corroborate the clinical efficacy of dietary therapy in managing inflammation and preventing carcinogenesis. This review seeks to explore the interrelationships among dietary antioxidants, dietary fiber, and the gut microbiome, with a particular focus on their potential implications in inflammation and cancer.

## 1. Introduction—The Human Microbiome

The investigation of the human microbiome has surged in recent decades, attracting great scientific curiosity and leading to interesting discoveries [1]. Originating from environmental microbiology, this field has transformed our understanding of organisms, conceiving eukaryotes as inseparable from their microbial companions within the human ecosystem [2]. The microbiome, denoting the collective genetic material of microorganisms inhabiting various human body sites, embodies a symbiotic relationship with its host, exerting profound influences on health [3]. The microbiota comprises bacteria, archaea, fungi, and various other microorganisms. The inclusion of viruses, phages, and mobile genetic elements within the microbiome remains a topic of debate, yet it underscores the complexity of microbial ecosystems [4]. The microbiota’s diversity contributes to ecosystem functionality and mutualistic interactions [5]. The symbiotic interplay between host and microbiome profoundly shapes physiological processes, augmenting host metabolism through an expansive production of enzymes, proteins, and metabolites [6]. Notably, the microbiome composition exhibits both anatomical and individual-specific variations, requiring a personalized approach to achieve its real role in health maintenance [7]. Dysbiosis is characterized by perturbations in microbial composition and often heralds the onset of disease, underscoring the delicate balance essential for health maintenance. While pathogens represent a minority, alterations in microbiome composition can precipitate their proliferation, emphasizing the intricate dynamics within microbial communities [8].

While the definitive definition of healthy microbiota remains elusive, interventions such as probiotics, prebiotics, and synbiotics offer promising avenues for modulating microbial ecosystems toward health promotion [9]. Central to this endeavor is the delineation of core microbiota, representing stable microbial communities associated with specific host genotypes or environments [10]. Identifying these core constituents facilitates the distinction between transient and persistent microbiota, guiding the design of robust experimental and statistical frameworks essential for therapeutic advancement in microbiome research [11].

Adaptation to the human microbiome involves mechanisms reflecting microorganisms’ preferred ecological niches [12]. Environmental parameters like temperature, pH, oxygen levels, pressure, osmolarity, and nutrient availability play pivotal roles in shaping microbial diversity and abundance across different bodily sites [7]. The dynamic composition of the human microbiota is governed by a complex interplay of **intrinsic and extrinsic factors**. **Intrinsic determinants** encompass the inherent characteristics of bodily environments, including physiological conditions that favor microbial growth [2]. Genetic predispositions, ethnic background, gender, and age further contribute to the unique microbial landscape of individuals, which tends to stabilize once microbial communities acclimate to their niches [13]. **Extrinsic influences**, such as dietary patterns, lifestyle choices, medication regimens, geographical location, climate variations, and seasonal fluctuations, exert considerable sway over microbial community dynamics [14] (Figure 1). 

Notably, the mode of childbirth significantly impacts early microbiota establishment, with neonates delivered vaginally or via cesarean section exhibiting distinct gut microbial profiles. Natural birth promotes the early colonization of the gut by microorganisms, resulting in higher gut bacterial counts compared with infants delivered by cesarean section [15]. On the other hand, bacteria were discovered in the placenta, umbilical cord, and amniotic fluid in full-term pregnancies, in accordance with relevant research [16]. This suggests that the fetus may not be completely sterile in utero and that the fetal intestinal flora may begin to colonize within the mother’s uterus [17]. Examinations of the umbilical cord blood flora from infants delivered by cesarean section found bacteria from the *Enterococcus*, *Streptococcus*, *Staphylococcus*, and *Propionibacterium genera* [18]. Furthermore, genetically labeled *Enterococcus faecium* was injected into the intestines of pregnant mice in animal experiments, and labeled *Enterococcus faecium* was found in the feces of young mice delivered by cesarean section, indicating the possibility of maternal microorganisms being transferred to the fetus in utero [19]. However, by age three, gut microbiota composition converges towards an adult-like state [20].

Antibiotic use, therapeutic interventions, and sanitation practices contribute to declining microbiota diversity [21] (Figure 1). Circadian disruption and antibiotic use further impact gut health by altering bacterial communities, reducing microbial diversity, and favoring the expansion of opportunistic pathogens [22]. Advancing age leads to physiological changes affecting gut microbiota composition and function, including alterations in digestive capabilities and nutrient absorption efficiency [23]. Diminished immune competence in older adults renders them more susceptible to pathogenic colonization, thereby modulating the core microbiome [24]. Notably, reductions in beneficial taxa like *Bifidobacterium* spp. in aging populations can precipitate malnutrition and heightened systemic inflammation [25].

The microbiome contributes to homeostasis across all bodily tissues [26]. Vertebrates exhibit symbiotic relationships with the diverse and intricate microbial communities inhabiting their gastrointestinal tracts [27]. This interaction between individuals and their microbiota is predominantly mutualistic, with the balance of gut microbial ecosystems (eubiosis) being fundamental [28]. Disruption of gut microbiota equilibrium can precipitate various chronic diseases [29]. Pathological alterations in the profile and functions of the microbiome are referred to as dysbiosis, which can instigate long-term inflammation and the production of carcinogenic compounds, potentially leading to neoplasia [30]. 

In summary, the human microbiome thrives within niches reflecting the body’s natural environment [31]. Disruptions to this equilibrium, driven by environmental perturbations, can induce shifts in microbial composition and diversity, potentially predisposing individuals to disease states [30]. This review aims to investigate the interactions among dietary antioxidants, dietary fiber, and the gut microbiome. We also aim to highlight their putative role in inflammation and cancer. 


**Literature Search Methodology**


Research articles and reviews retrieved from diverse global databases were systematically examined using specified phrases and keywords for this investigation. The keywords employed included antioxidants, dietary fiber, gut microbiome, inflammation, cancer, vitamins, flavonoids, and other nutritional-derived antioxidants. This review encompassed literature spanning from 2005 to the present.

The topics addressed to achieve this study’s objectives included altering the population dynamics of the microbiome, microbiota-mediated biosynthesis of metabolites, dietary influence on the microbiome, the impact of fiber on the gut microbiome, and the regulatory role of diet–microbiota interactions in inflammatory and gastrointestinal diseases within the intestinal milieu.

### 1.1. The Oral Cavity

The human oral cavity hosts a diverse microbiota comprising bacteria, fungi, viruses, and protozoa, colonizing both the hard surfaces of teeth and the soft tissues of the oral mucosa [32]. Serving as a primary gateway to the body, the oral microbiome holds significance in health and disease, influencing systemic immunity and physiological processes [33].

Within the oral cavity, microbial activities like nitrate metabolism yield nitric oxide, a compound pivotal for antimicrobial defense and vascular function [34]. Perturbations in oral microbiota composition have been implicated in various chronic ailments, including endocarditis, osteoporosis, and rheumatoid arthritis [35]. Additionally, oral health status correlates with the onset and progression of non-communicable diseases such as obesity, diabetes, and certain cancers, as well as neuropsychiatric disorders [36]. Recognizing the potential of the oral microbiome as a diagnostic tool for disease risk assessment parallels the evolving understanding of its systemic implications [37]. Analogous to investigations into the gut microbiome, contemporary research endeavors seek a comprehensive comprehension of oral microbiome functions and interactions within the human body [32]. Prospective inquiries are poised to elucidate strategies for restoring oral microbiota equilibrium and promoting systemic health [38] (Figure 2).

### 1.2. The Stomach

The stomach was previously considered sterile because of hostile conditions like acidity and mucus thickness; however, the stomach’s microbial landscape has been redefined following the discovery of *Helicobacter pylori* [39]. Despite challenges in diagnostic methodologies, recent insights have unveiled a diverse gastric microbiome predominantly composed of the phyla *Proteobacteria, Firmicutes* (formerly known as *Bacillota*), *Bacteroidetes* (formerly known as *Bacteroidota*), *Actinobacteria,* and *Fusobacteria* within the gastric mucosa [40].

Diverging from mucosal populations, gastric juice hosts a distinct microbial community primarily consisting of *Firmicutes, Actinobacteria*, and *Bacteroidetes,* with *Proteobacteria* and *Firmicutes* prevailing in mucosal environments [40]. Transient colonization by oral and duodenal bacteria like *Veillonella, Lactobacillus*, and *Clostridium* further underscores the dynamic nature of gastric microbiota [41].

*Helicobacter pylori*, commonly found in individuals with infections, induces substantial changes in the gastric environment, which may significantly influence the indigenous microbial ecosystems residing therein [42]. These shifts correlate with heightened risks of gastric cancer development, emphasizing the microbial influence on disease pathogenesis [43]. Remarkably, *Helicobacter pylori* eradication has been associated with enhanced microbial diversity within the stomach [44] (Figure 2). Stomach mucosa inflammation, medication use, dietary behaviors, and, especially, H. pylori colonization impact the dynamic makeup of the stomach microbiota at the general level. A limited number of studies, mostly animal model studies, have examined the impact of nutrition on gastric microbiota, although several studies have reported the influence of diet on gut microbiota composition in humans [45]. In comparison with mice fed a purified diet (refined food), an in vivo investigation revealed higher amounts of total aerobes, total anaerobes, and lactobacilli in the stomach of mice fed a non-purified diet (natural source-derived food). Lower levels of Toll-like receptor 2 (TLR-2) mRNA in the stomach were correlated with these increases [46]. Given that the diversity of the gastric microbiota depends on gastric acid secretion, it is not surprising that long-term usage of H2-antagonists and proton pump inhibitors (PPIs) also impacts the composition of the gastric microbiota in cases of atrophic gastritis. A pH of more than 3.8 in the stomach leads to bacterial overgrowth [46]. PPI therapy patients have considerably higher levels of oro-pharyngeal-like and fecal-like bacteria than H2-antagonist patients and untreated control participants [47]. The ecological impacts of antibiotics on the gut microbiota are well-documented. Investigations on animals revealed that the administration of penicillin decreases the number of Lactobacilli and encourages the colonization of the stomach epithelium by yeast [48].

While the intricacies of *H. pylori* interactions with commensal bacteria remain elusive, its profound impact on the gastric microbiome offers encouraging prospects for therapeutic modulation, hinting at avenues to mitigate disease progression [49].

### 1.3. Human Gut Intestinal Microbiota

The human gastrointestinal tract is inhabited by trillions of bacteria spanning diverse kingdoms of life, collectively known as the gut microbiota. These microbes play essential roles in host growth, physiology, and health maintenance [28]. The gut microbiota’s intricate ecosystem communicates internally and with the host, influencing vital biological functions such as juvenile growth regulation, immune system maturation, and defense against pathogens [50]. Additionally, it contributes significantly to carbohydrate and lipid metabolism, crucial for maintaining the host’s energy balance [51]. Notably, the intestinal mucosal layer not only serves as a carbon source but also facilitates microbial adherence [52].

The gastrointestinal tract, especially, the large intestine, houses a densely populated and diverse microbial community, with a bacterial-to-host cell ratio approximating 1:1 [53]. Predominantly, commensal bacteria inhabit the colon, while lesser populations reside in the stomach and small intestine. Firmicutes and Bacteroides phyla constitute the majority of gut microbiota, with *Actinobacteria, Proteobacteria, Fusobacteria*, and *Verrucomicrobia* also present [25]. *Firmicutes*, notably, *Bacillus*, *Lactobacillus*, *Enterococcus*, *Clostridium*, and *Ruminococcus genera*, encompass approximately 200 distinct genera within this phylum [54].

Although some *Firmicutes* species, like *Staphylococcus aureus* and *Clostridium perfringens*, may pose health risks when overgrown, others such as Bifidobacterium demonstrate beneficial effects [3]. Similarly, *Bacteroides* and *Prevotella* are predominant genera within the *Bacteroidota phylum* [55]. The less prevalent Actinobacteria phylum is typified by *Bifidobacterium*, known for its positive influence on health, while *Proteobacteria harbor* pathogens like *Enterobacter*, *Helicobacter*, *Shigella*, *Salmonella*, and *Escherichia coli* [56] (Figure 2).

The large intestine, characterized by slow flow rates and a pH ranging from mildly acidic to neutral, hosts the largest microbial community, primarily composed of obligate anaerobes [25]. Within this environment, distinct microenvironments exist. The epithelial surface and inner mucin layer typically have minimal colonization during health, while the diffuse mucin layer hosts specialist colonizers like *Akkermansia muciniphila*. In the liquid phase of the gut lumen, a diverse microbial population thrives, including primary colonizers like *Ruminococcus *spp., influenced by dietary fibers [57].

Conversely, the small intestine, with its short transit time of approximately 3–5 h, poses challenges for microbial colonization because of high bile concentrations with antimicrobial properties [58]. Molecular analysis indicates that the jejunal and ileal components mainly harbor facultative anaerobes, such as *Proteobacteria, Bacteroides, Streptococci, Lactobacilli*, and *Enterococci species* [59].

Gut bacteria play a vital role in regulating digestion and metabolizing nutrients like short-chain fatty acids (SCFAs), bile acids, and amino acids, contributing to host energy extraction and metabolic efficiency [60]. They also aid in maintaining intestinal epithelial integrity, thereby preventing the invasion of pathogenic bacteria and supporting immune function [61]. Gut bacteria metabolize dietary and endogenous substrates into metabolites that facilitate communication with the host’s peripheral organs and tissues [62]. Alterations in gut bacterial composition have been associated with specific diseases, suggesting dietary interventions as potential therapeutic avenues [63]. Dietary fiber, for instance, confers resistance against colonization by multidrug-resistant bacteria and enhances the abundance of bacteria involved in SCFA production [64]. The intestinal microbiota and the gut mucosal immune system maintain a symbiotic relationship, and disruptions to this interaction can lead to disease [65]. Western dietary patterns, characterized by low microbiota-accessible carbohydrates, alter microbiota composition and functionality compared with non-Westernized populations with high-fiber diets. Immunological dysregulation resulting from host–microbiota interactions underpins various inflammatory disorders [66]. The composition and metabolic activity of the gut microbiome are influenced by substrate availability and external factors like diet [67]. Microbially synthesized metabolites facilitate communication among metabolic, immune, and neuroendocrine systems, impacting overall host health [68].

In addition to their digestive functions, commensal gut bacteria occupy niches that suppress pathogenic colonization [69]. Disruption of this balance, leading to increased gut permeability, allows opportunistic pathogens to colonize, potentially resulting in dysregulated metabolite production and various diseases, including inflammation, allergies, and autoimmune disorders [70].

Opportunistic pathogens, such as *Clostridium difficile*, are normally present in the gut microbiota; nonetheless, they can become pathogenic when the healthy microbiome state is disturbed, leading to conditions such as *Clostridium difficile* infection (CDI) [71]. *Clostridium difficile*, also known as *C. difficile*, is a significant healthcare pathogen and the primary cause of infections associated with medical treatment. However, in the last 20 years, the epidemiology of CDI has evolved following the appearance of a hypervirulent clone called NAP1/027/BI, which was linked to significant global outbreaks in the early 2000s [72]. The host microbiota and its related metabolites, the host immune system, and antimicrobial agents all have an impact on the life cycle of *Clostridium difficile*. Large clostridial toxins, toxins A (TcdA) and B (TcdB), and, in certain bacterial strains, the binary toxin CDT are the main mediators of inflammation in *C. difficile* infection (CDI) [73]. Toxins set off a complicated succession of host cellular reactions that result in diarrhea, inflammation, and tissue necrosis—the three main signs and symptoms of traumatic CDI [74]. It is unclear what exactly causes the pandemic of certain strains of *C. difficile*. CDI-associated symptoms range from diarrhea to pseudomembranous colitis and sepsis. The dominant gut microbiota in a healthy state typically prevents the overgrowth of *Clostridium difficile*, illustrating the concept of colonization resistance, particularly evident in antibiotic-associated diarrhea cases compared with other pathogens like *Salmonella species* [75]. Nevertheless, fecal microbiota transplantation has demonstrated effectiveness in treating recurrent infections.

The gut microbiota composition undergoes dynamic changes throughout life stages, notably during birth, weaning, and aging [76]. Facultative anaerobes initially colonize the gut post-birth, fostering an anaerobic milieu conducive to obligate anaerobe growth, including *Bifidobacterium* and *Bacteroides species* [77]. Neonates born vaginally acquire maternal vaginal and fecal microbiota, while those born via Cesarean section initially harbor skin-associated microbiota [78]. Breastfed infants tend to exhibit greater abundance and diversity of *Bifidobacterium* spp. compared with their formula-fed counterparts until weaning, after which gut microbiota diversity increases with solid food introduction [79]. There is a firmly established correlation between diet and CRC, and dietary patterns likely have the greatest impact on CRC. A contributing cause to the rising rate of CRC in Western countries may be the prevalent low dietary intake of fiber [80]. Several epidemiologic studies have demonstrated a negative correlation between daily fiber consumption and CRC risk. Dietary fibers impact survivors of CRC’s long-term prognosis. Numerous pathogenetic mechanisms have been proposed [81]. Fibers have the potential to disrupt bile acid metabolism, which could accelerate the development of colon cancer. Moreover, veggies are frequently high in fiber as well as rich in antioxidants such as phytoestrogens, polyphenols, and resveratrol [81]. Sporadic types of CRC are also caused by certain genetic changes that are linked to hereditary forms. Wnt/β-catenin, TGF-β receptor, Notch, and Hedgehog pathways are specifically engaged [82]. 

During aging, reduced microbiota diversity, diminished *Bifidobacteria*, and increased *Enterobacteriaceae* levels are observed [23]. Environmental factors, anatomical location, mode of delivery, milk feeding method, weaning, age, diet, and antibiotic usage profoundly influence gut microbiota composition and variation.

Dominant gut microbiota members play pivotal roles in digestion regulation throughout the gastrointestinal tract, metabolizing nutrients like SCFAs, bile acids, and amino acids, thereby enhancing host energy harvesting and metabolic efficiency [60]. Additionally, these commensals fortify the intestinal epithelium, preventing pathogenic invasion and maintaining mucosal integrity [83].

Dysbiosis, or microbial imbalance, increases gut permeability, permitting opportunistic pathogens to colonize and proliferate, potentially leading to disease states [30]. Pathogens like *Clostridioides difficile*, which are usually part of the healthy microbiota, can become pathogenic under dysbiotic conditions, causing symptoms ranging from diarrhea to severe colitis [84]. Inflammatory bowel disease (IBD) exemplifies gut dysbiosis-associated conditions, characterized by inflammation along the gastrointestinal tract, potentially attributed to reductions in beneficial *Firmicutes* species like *Faecalibacterium prausnitzii* and *Roseburia *spp. [85].

Beyond IBD, dysbiosis is implicated in irritable bowel syndrome (IBS), celiac disease, and CRC [29]. Reductions in *Lactobacillus* and *Firmicutes* and increases in *Firmicutes/Bacteroidetes* ratios have been observed in IBS patients, indicative of altered microbial composition associated with disease pathogenesis [86]. Understanding the role of microbial activities in disease etiology offers avenues for therapeutic intervention and underscores the critical influence of the gut microbiome on human health and disease [67].

The human intestinal microbiome plays crucial roles in digesting indigestible food, synthesizing essential nutrients such as vitamins, metabolizing various compounds, modulating immune responses, promoting epithelial cell renewal, maintaining mucosal integrity, and producing antimicrobial substances [12]. The microbiome, encompassing bacteria, viruses, fungi, and other microorganisms, profoundly impacts human health, and alterations in microbiota composition can lead to various diseases. Dysbiosis in the intestinal microbiome is associated with environmental risk factors that contribute to the onset and progression of CRC [50]. Emerging evidence suggests that changes in gut microbiota occur early in CRC development and can serve as predictive markers for identifying individuals at risk of colorectal adenoma, a precursor lesion to CRC [87]. Thus, microbiota alterations hold promise as biomarkers for early CRC diagnosis [88]. 

The intricate relationship between the gastrointestinal (GI) microbiome and the progression of chronic non-communicable diseases underscores the significance of developing strategies to modulate the GI microbiota for promoting human health.

## 2. Changing the Population Dynamics of the Microbiome

The population dynamics of the human microbiome are highly individualized and subject to continual variation influenced by factors like age, dietary patterns, host genetics, and medication usage [89]. Despite this inherent variability, specific microbiome signatures have been correlated with various diseases, as evidenced by disparities between patient cohorts and healthy counterparts [90]. 

A diminished microbial diversity within the human microbiome is notably linked to various diseases. However, alterations in microbiome composition can exhibit distinct patterns across different populations [91]. There is conflicting evidence regarding the role of the *Firmicutes/Bacteroidetes (F/B)* ratio as a biomarker for obesity, as recent data have shown no association between the *F/B* ratio and obesity. On the one hand, the *F/B* ratio, a pivotal parameter reflecting microbiome diversity, particularly in the gut, has garnered significant attention [92]. On the other hand, Houtman and his colleagues suggested that there is insufficient evidence to substantiate the energy harvesting theory about the association between infant and childhood obesity and gut microbiota. Unlike what was anticipated, they only identified a small number of significant correlations between zBMI and the *F/B* ratio or SCFA producers, and these correlations were abolished when the Benjamini–Hochberg technique was applied [93]. However, elevated *F/B* ratios were noted in obese populations and were consistently associated with obesity across diverse studies [92]. Conversely, reduced *F/B* ratios were evident in patients with IBD, especially in Crohn’s disease (CD) and ulcerative colitis (UC) [94]. Notably, decreased levels of *Firmicutes*, particularly the *Faecalibacterium genus*, were observed in patients with major depressive disorder, bipolar disorder, and chronic fatigue syndrome [95]. 

In the rapidly evolving field of nutrition and metabolism research [96], we present pivotal findings that offer insights for guiding future investigations using Next-Generation Sequencing (NGS) of the microbiome and implementation research. We completed a meticulous examination of individual diet–health relationships as presented by Zeevi et al., who identified significant variability in human dietary metabolic processing and physiological responses [97]. Notably, they found considerable inter-individual variation in postprandial glycemic responses to identical meals and composite foods like pizza. They developed a machine learning algorithm integrating microbiome NGS data and other factors to predict these responses, revealing the potential for personalized dietary recommendations based on microbiome insights obtained by the NGS methodology.

Additionally, next-generation probiotics (NGPs) are gaining popularity as therapeutic agents following advancements in high-throughput DNA sequencing and molecular analysis technologies. However, NGPs face greater complexity in their functioning within the gastrointestinal tract compared with traditional probiotics when encountering hostile conditions [96]. NGPs offer several advantages over conventional probiotics [97], such as personalized treatments, the integration of synthetic biology and gene editing, the potential for combination therapies, targeted delivery, and application in therapeutic settings.

NGPs are believed to modulate the gut microbiota, potentially preventing viral and neurodegenerative conditions, reducing oxidative stress, and influencing inflammatory pathways [98]. Prebiotics play a critical role in enhancing the efficacy of both traditional probiotics and NGPs. Defined as substrates hosting microorganisms that are preferentially utilized to confer health benefits, prebiotics contribute to several health-promoting mechanisms including the following: Modulation of gut microbiota composition and production of beneficial microbial metabolites such as SCFAs and tryptophan [99].Direct stimulation of probiotic growth and fermentation processes.Provision of encapsulating materials for probiotics.

These interactions underscore the symbiotic relationship between prebiotics and NGPs, highlighting their combined potential to improve gut health and therapeutic outcomes.

### 2.1. Probiotics and Their Role in Gut Microbiota—Manipulating the Microbiome Composition through Probiotic Supplementation

Probiotic bacteria exert influence not only on the intestinal microbiota within the large intestine but also on other organs by modulating immunological parameters, intestinal permeability, bacterial translocation, and providing bioactive or regulatory metabolites [98]. According to the World Health Organization (WHO), probiotics are defined as “live microorganisms that, when administered in adequate amounts, confer a health benefit on the host” [99]. The majority of health benefits attributed to probiotic microorganisms are associated with the gastrointestinal (GI) tract, either directly or indirectly through modulation of the immune system [98]. While probiotics are typically administered orally, interactions with the host’s indigenous microbiota or immunocompetent cells within the intestinal lumen largely determine the mechanisms and efficacy of probiotic effects [100]. The gut, or the gut-associated lymphoid tissue (GALT), serves as the largest immunologically competent organ, and the growth and composition of the resident microbiota play crucial roles in the maturation and optimal development of the immune system from birth [100].

Probiotics exhibit the potential to ameliorate or prevent gut inflammation and various intestinal or systemic disease phenotypes by modulating the composition of the gut microbiome and conferring beneficial functionalities to gut microbial communities [101]. Several strains of probiotic bacteria have been identified to modulate the microbiota of the small intestine and/or restrict pathogen colonization in the gut, thus impeding the translocation of pathogenic bacteria across the intestinal barrier and subsequent infection of other organs [98]. The precise mechanisms underlying these effects remain elusive but may include reduced intestinal pH, production of bactericidal compounds (e.g., organic acids, H_2_O_2_, and bacteriocins), agglutination of pathogenic bacteria, and reinforcement of the intestinal mucosal barrier function [102]. Probiotics may also compete for microbial fermentation substrates or receptors on mucosal surfaces and release gut-protective compounds such as arginine, glutamine, SCFAs, and conjugated linoleic acid (CLA) [103]. Beyond their immunomodulatory properties, probiotics have been investigated for their potential utility in the prevention or management of diarrheal diseases or IBD [104] (Table 1).

Probiotics have been shown to substantially reduce antibiotic-associated diarrhea, traveler’s diarrhea, and diarrhea of various etiologies [105]. The most well-documented benefit of probiotics is their efficacy in preventing infections or alleviating symptoms associated with immunostimulatory bacteria or yeasts [100]. The administration of *Lacticaseibacillus rhamnosus* (formerly known as *Lactobacillus rhamnosus*) daily for two months was associated with a reduced incidence of diarrhea and gastrointestinal and respiratory tract infections [106]. 

Many studies have shown that probiotic supplementation before and during antibiotic therapy reduces the duration, frequency, and severity of antibiotic-associated diarrhea episodes [107]. Chemotherapy and radiotherapy often disrupt the immune system and intestinal microbiota, leading to diarrhea and increased *Candida albicans* colonization in the gastrointestinal tract and other organs. The administration of probiotic microorganisms before and during chemotherapy has shown promise in mitigating these side effects in animal models [108]. Similarly, in individuals with lactose malabsorption, fermented milk products improve lactose digestion and alleviate intolerance symptoms [109]. This effect is largely attributed to the presence of microbial galactosidase in fermented milk products containing live bacteria, which survives passage through the stomach and facilitates lactose hydrolysis in the small intestine [109].

The preceding studies underscore the significant association between reduced levels of specific microbial phyla or genera in the microbiome and the onset and progression of diseases. Therefore, rectifying this imbalance by modulating the microbiome is crucial for mitigating associated disorders and fostering health. One approach to achieve this is by exogenously administering beneficial bacteria, such as probiotics, to restore microbiome equilibrium. Previous investigations have explored this strategy and yielded promising outcomes in certain contexts [110].

Beyond conventional probiotic strains like *Lactobacillus* and *Bifidobacterium*, the versatile *Akkermansia muciniphila* has emerged as a promising candidate for treating cardiometabolic diseases such as obesity and diabetes [111]. Given its clear inverse correlation with such pathologies, a randomized, double-blind, placebo-controlled study demonstrated the safety and efficacy of *Akkermansia muciniphila* supplementation in improving insulin sensitivity, reducing cholesterol and body weight, and ameliorating markers of liver dysfunction in overweight or obese individuals [112] (Table 1).

**Table 1 ijms-25-08250-t001:** Probiotics and their effects on the gut microbiota.

Probiotics	Effects	References
	1. Probiotics can influence the composition of the gut microbiome and have positive effects on gut microbial populations, which may help to reduce or prevent gut inflammation and other phenotypes of intestinal or systemic disease.	[101]
	2. The specific processes that underlie these actions are still unknown, but they might involve the following: lowering the intestinal pH; producing bactericidal substances (such organic acids, H_2_O_2_, and bacteriocins); agglutinating harmful bacteria; and bolstering the effectiveness of the intestinal mucosal barrier.	[102]
	3. Probiotics can compete with other microorganisms for substrates or receptors on mucosal surfaces, leading to the production of chemicals that are beneficial to the gut, including glutamine, arginine, SCFAs, and conjugated linoleic acid (CLA).	[103]
	4. Daily administration of *Lacticaseibacillus rhamnosus*, formerly known as *Lactobacillus rhamnosus*, for two months was linked to a decreased occurrence of respiratory tract infections, diarrhea, and gastrointestinal disorders.	[106]
	5. Probiotic bacteria administered prior to and during chemotherapy have demonstrated potential in reducing these adverse effects in animal models.	[108]
	6. In addition to traditional probiotic strains like *Lactobacillus* and *Bifidobacterium*, the adaptable *Akkermansia muciniphila* is gaining interest as a possible treatment option for cardiometabolic disorders like diabetes and obesity.	[111]
	7. After administering a combination of *L. plantarum ZDY2013* and *B. bifidum WBIN03* to mice with dextran sodium sulfate (DSS)-induced UC, Wang et al. showed the downregulation of pro-inflammatory cytokines and the overexpression of antioxidant factors.	[113]

SCFAs, short-chain fatty acids; CLA, conjugated linoleic acid; DSS, dextran sodium sulfate.

Furthermore, probiotic interventions have been explored for managing IBD. Wang et al. demonstrated the downregulation of pro-inflammatory cytokines and upregulation of antioxidant factors in mice with dextran sodium sulfate (DSS)-induced UC following administration of a mixture of *L. plantarum* ZDY2013 and *B. bifidum* WBIN03 [113]. Despite promising findings in animal models, translating these effects to clinical settings remains challenging [114]. 

In summary, probiotic supplementation offers a promising avenue for modulating microbiome composition to address various diseases, yet further research is needed to elucidate mechanisms and optimize therapeutic outcomes.

### 2.2. Modulating Microbiome Composition through Prebiotic Supplementation

An alternative approach to exogenously supplementing probiotics is the administration of prebiotics, which are nondigestible substrates utilized by members of the host microbiome to confer health benefits [115]. Prebiotics typically comprise oligosaccharides that selectively stimulate the growth of specific bacterial species already present in the microbiome [116]. By promoting the expansion of beneficial microbes, prebiotics offer a means to remodel the microbiome from a diseased state to a healthier state [117]. Common prebiotics include fructo-oligosaccharides (FOSs) derived from inulins, galacto-oligosaccharides (GOSs), xylo-oligosaccharides (XOSs), and lactulose. While FOSs, GOSs, and XOSs have demonstrated a capacity to enhance *Bifidobacterium* proliferation in the human gut, the effects on other bacterial genera remain variable because of variations in intervention doses and durations [118]. The application of prebiotics extends to cancer therapy as well [119]. Notably, preclinical studies have demonstrated that responders to immune checkpoint blockers harbor a higher abundance of beneficial bacteria such as *Bifidobacterium*, *Akkermansia*, *Ruminococcaceae*, and *Faecalibacterium* in their gut microbiome compared with non-responders [120]. Han et al. revealed that the oral administration of inulin gel in mice led to the expansion of beneficial bacteria, including *Akkermansia*, *Lactobacillus*, and *Roseburia.* This elicited a T cell response that synergized with α-PD-1 for enhanced antitumor effects [121] (Table 2). 

In conclusion, probiotics and prebiotics aim to enhance the population of beneficial bacteria in the intestinal lumen post-consumption [118]. Probiotics introduce health-promoting bacteria, while prebiotics supply fermentable carbohydrates that selectively stimulate the growth of beneficial indigenous intestinal bacteria [116]. Prebiotic polysaccharides, categorized as dietary fibers, are fermented by the large intestinal microbiota, promoting the health of the intestinal mucosa, increasing biomass and fecal weight, regulating defecation frequency, and improving overall gut health [122].

## 3. Biosynthesis of Metabolites by Microbiota

Metabolites produced by the microbiota play a pivotal role in host–microbe interactions, often exerting greater influence than the microbes themselves [62]. Despite this significance, the specific metabolites and biosynthesis pathways underlying these interactions remain largely elusive [123]. Various studies have demonstrated that microbes engage in host modulation through metabolite-mediated signaling pathways [124]. For example, SCFAs such as acetate, butyrate, and propionate, derived from dietary fibers in the colon, are known to influence the differentiation and accumulation of regulatory T cells (Treg cells) by activating G-protein coupled receptors [125]. Consequently, activated Treg cells produce anti-inflammatory factors like IL-10, which are believed to mitigate gut inflammatory conditions such as IBD [126]. The connection between microbiome composition and metabolites is often indirect because of the functional redundancy of metabolic pathways, suggesting potential species interchangeability [127]. Therefore, differences in microbial composition may not necessarily reflect functional metabolite variations. In fact, attempts to predict phylogeny from metabolomic data have proven challenging, underscoring the complexity of linking microbiome and metabolome data. 

Thus, identifying disease-associated metabolites can offer more straightforward insights, facilitating the implementation of synthetic approaches for therapeutic interventions [128].

A case in point is the study recently published by Zhang et al. [129]. Previous research has demonstrated that Lactobacillus species play a significant role in mitigating CRC in murine models, yet the precise mechanisms remain incompletely understood. Zhang et al.’s studies revealed that indole-3-lactic acid facilitated the production of IL12a in dendritic cells by augmenting H3K27ac binding at enhancer sites within the IL12a gene locus, thereby promoting the priming of CD8+ T cell-mediated immune responses against tumor progression. Additionally, indole-3-lactic acid transcriptionally repressed Saa3 expression, which is implicated in cholesterol metabolism in CD8+ T cells, by modulating chromatin accessibility. This modulation enhanced the functional capacity of tumor-infiltrating CD8+ T cells. Additionally, Zhang et al. observed that the administration of the probiotic strain *Lactobacillus plantarum* L168, along with its metabolite indole-3-lactic acid, alleviated intestinal inflammation, suppressed tumor growth, and mitigated gut dysbiosis. Their study provided novel insights into the epigenetic mechanisms underlying probiotic-mediated anti-tumor immunity. Moreover, they underscored the therapeutic potential of *L. plantarum* L168 and indole-3-lactic acid in the development of therapeutic interventions for patients afflicted with CRC.

## 4. Diet and Microbiome

Diet plays a significant role in human health and disease, particularly in the development of metabolic disorders such as obesity, diabetes, and hypertension [130]. These conditions often stem from underlying inflammation, a tightly regulated immune response where specialized immune and non-immune cells release inflammatory molecules in response to infection or tissue damage [131]. This cascade of inflammation recruits immune cells to contain and eliminate the threat, resolving tissue damage. However, if inflammation persists unchecked, it can lead to chronic conditions and pathology. Many metabolic and immune disorders are now understood to be linked to dysregulated inflammatory responses [131].

While the relationship among the immune system, inflammation, and diet in metabolic diseases is recognized, the specific impact of dietary factors on inflammation remains poorly understood [132]. Likewise, our understanding of how diet interacts with gut microbiota to regulate immune function is still developing [133]. It is crucial to consider both gut and broader host physiology in how nutrients interact with the body and its resident gastrointestinal microbes [134]. 

Diet significantly influences the composition, diversity, and richness of the gut microbiota [135]. Various components of the diet exert time-dependent effects on gut bacterial ecosystems. As diet is the most readily modifiable factor, it represents a straightforward therapeutic intervention [136]. *Prevotella*, *Bacteroides*, and *Ruminococcus* are among the prevalent enterotypes identified in studies of gut microbial populations [137]. 

Notably, the *Prevotella/Bacteroides* ratio varies across populations with distinct dietary habits, indicating the influence of long-term dietary differences, such as fiber-rich diets in non-Westernized populations and meat-rich diets in Westernized populations [138]. 

Important epigenetic processes are regulated by the microbial metabolism of various food ingredients, which ultimately affects host health. The gut microbiome is affected by diet-mediated changes that control the substrates that are accessible for epigenetic modifications such as acetylation, histone methylation, and/or DNA methylation. Furthermore, the production of microbial metabolites like butyrate suppresses the activity of key enzymes involved in epigenetic modification, such as histone deacetylases (HDACs) [139,140]. In turn, the host’s epigenome affects the makeup of gut microbes. As a result, these three elements interact in intricate ways [141].

## 5. The Role of Fiber on the Gut Microbiome

Dietary fibers are characterized as ingestible carbohydrate polymers comprising three or more monosaccharide units resistant to endogenous digestive enzymes, thereby escaping metabolism and absorption in the small intestine [142]. They are found in consumable carbohydrate polymers present in natural foods like fruits, vegetables, legumes, and cereals; in raw edible materials undergoing physical, enzymatic, or chemical degradation; and in synthetic carbohydrate polymers with established biological benefits [142]. The WHO advocates a daily intake of at least 25 g of fiber [143]. Recommendations from various food and health organizations emphasize diets rich in vegetables, fruits, and whole-grain cereals to meet these guidelines. For adults, most countries suggest a daily intake of 25–35 g of dietary fiber, with recommendations ranging from 18 to 38 g per day [144]. 

Dietary fibers encompass various polysaccharides, including non-starch polysaccharides (NSPs), resistant starch (RS), and resistant oligosaccharides (ROs), categorized as either soluble or insoluble [142]. Insoluble fibers, such as cellulose and hemicellulose, resist digestion in the small intestine and proceed to the colonic region, where they contribute to fecal bulk without significant fermentation by gut bacteria [145]. In contrast, most soluble NSPs, particularly high molecular weight polymers like guar gum, certain pectins, β-glucans, and psyllium, exhibit viscosity and gel-forming properties in the intestinal tract, delaying glucose and lipid absorption and impacting postprandial metabolism [146].

Soluble fibers, while not primarily contributing to fecal bulking, are metabolized by gut bacteria to produce SCFAs [147]. SCFAs, predominantly acetate, propionate, and butyrate, play crucial roles in regulating host metabolism, immune function, and cell growth. Predominantly present in the cecum and proximal colon, SCFAs serve as energy substrates for colonocytes, particularly butyrate, and also exert systemic effects by entering the portal circulation and influencing hepatocytes and peripheral tissues [125]. Despite their relatively low levels in peripheral circulation, SCFAs serve as signaling molecules, modulating various physiological functions within the host [125]. Soluble fibers prolong intestinal transit time and undergo fermentation in the colon, producing gases. Insoluble fibers can either pass through the intestine inertly, augmenting fecal bulk, or undergo fermentation, hastening food passage through the digestive tract [148] (Table 3).

Soluble fibers bind to bile acids in the small intestine, impeding their absorption and consequently lowering blood cholesterol levels [149]. Additionally, they inhibit sugar absorption, normalize blood cholesterol, and generate SCFAs in the colon through fermentation, exerting diverse physiological effects [150]. Insoluble fibers are associated with reduced diabetes risk, but the precise mechanisms remain unclear [151].

*Prevotella* plays a crucial role in fostering a healthy gut microbiota [25]. Comparative analyses of gut bacteria from children in rural and urban communities have highlighted the impact of nutrition on the microbiome. Children from rural communities exhibited increased populations of *Prevotella* and *Xylanibacter genera*, associated with higher fecal SCFA levels, indicative of their ability to metabolize complex carbohydrates [152]. These microbiota variations transcend racial differences, as urbanized children from rural areas exhibited microbiomes resembling those of Italian children, emphasizing the significant influence of nutrition independent of host genetics [152].

Dietary fibers and SCFAs stimulate mucus production and secretion, with both acetate and butyrate regulating mucus production and release [153]. Acetate- and propionate-producing bacteria, such as *Bacteroides thetaiotaomicron*, promote goblet cell development and the expression of mucin-related genes [154]. Conversely, *Faecalibacterium prausnitzii*, a butyrate producer, reduces acetate’s influence on mucus production and inhibits mucus overproduction, maintaining the integrity of the gut epithelium [155]. Additionally, dietary fibers mechanically enhance mucus secretion by the intestinal epithelium [156].

**Table 3 ijms-25-08250-t003:** The effects of fiber on the gut microbiome.

Fibers	Effects	References
Soluble fibers	They are not the main cause of fecal bulking, but gut bacteria metabolize them to create SCFAs.	[147]
	SCFAs penetrate the portal circulation and affect hepatocytes and peripheral tissues in addition to providing colonocytes with energy substrates, especially butyrate, which has systemic effects.	[125]
	The intestinal transit time is extended by soluble fibers, which also ferment in the colon to release gasses.	[148]
	Soluble fibers bind to bile acids in the small intestine, impeding their absorption and consequently lowering blood cholesterol levels.	[149]
	Mucus secretion and production are induced by dietary fibers and SCFAs, whereas the release of mucus is controlled by butyrate and acetate.	[153]
Insoluble fibers	They can either go through fermentation, which speeds up food passage through the digestive tract, or they can pass through the colon inertly, increasing the weight of the feces.	[156]
	They are linked to a lower risk of diabetes, but the exact mechanisms are still unknown.	[157]

SCFAs, short-chain fatty acids.

Long-term dietary fiber deficiency correlates with an increase in mucin-degrading bacteria like *Akkermansia muciniphila*, compromising the mucus barrier. In the absence of dietary fibers, certain gut bacteria adapt their metabolism to utilize mucin glycans by upregulating mucin-degrading enzyme gene expression [156].

Feeding mice a Western diet (low in fiber) increased the permeability of the inner mucosal layer and slowed mucus turnover, potentially heightening susceptibility to infections [157]. In obese mice, supplementation with a low dose of inulin (1%) or *Bifidobacterium longum* restored mucus defects. Inulin supplementation restored inner mucosal layer permeability, while *Bifidobacterium longum* supplementation normalized the mucus turnover rate, suggesting independent mechanisms for each criterion [158]. High inulin intake (20%) prevented microbial invasion, improved intestinal health, and alleviated low-grade inflammation in obese mice [159].

Colonocytes metabolize butyrate aerobically via beta-oxidation, creating an anaerobic environment in the gut [160]. Reduced oxygen levels decrease butyrate-producing bacteria, promoting the growth of proteobacteria like *Escherichia coli* and *Salmonella enterica serovar Typhimurium* [161]. This feedforward loop elucidates the diseases associated with low-fiber diets and provides a molecular basis for decreased microbiota richness observed in humans and mice consuming low-fiber diets [162] (Table 3).

In conclusion, dietary fibers serve as a significant energy source for bacteria inhabiting the cecum and colon. Under specific intestinal conditions, anaerobic bacteria activate enzymatic machinery and metabolic pathways to metabolize complex carbohydrates, yielding metabolites such as SCFAs [67]. SCFAs, primarily acetate, propionate, and butyrate, are vital for maintaining intestinal mucosal health and are not readily available in the diet. The colonic microbiota ferments dietary fiber to meet metabolic requirements, leading to the production of luminal SCFAs [163]. It was previously demonstrated that colonocytes prefer to utilize butyrate over glucose as an energy source, highlighting the symbiotic relationship between colonic microorganisms and mucosal health [164].

## 6. Regulation of Inflammatory and Gastrointestinal Disease by Diet–Microbiota Interactions in the Bowel

Diet plays a pivotal role in modulating inflammation, both directly through interaction with immune cells and receptors and indirectly by influencing the gut microbiota [165]. Various components of the diet, including fats, proteins, carbohydrates, and micronutrients, serve as substrates for the gut microbiota [67]. Notably, dietary fibers, as non-digestible carbohydrates, serve as a significant energy source for gut microbes [166]. Changes in dietary fiber intake can swiftly alter the composition and function of the gut microbiota [67]. Microbes such as *Roseburia* and *Faecalibacterium*, which thrive in fiber-rich environments, are associated with lower levels of intestinal inflammatory markers [167].

Dietary fibers also contribute to local immune homeostasis in the gut, with high-fiber diets showing protective effects in murine models of IBD [165]. Moreover, increased fiber intake is linked to a reduced incidence of Crohn’s disease, particularly when derived from fruits [168]. These effects may be attributed to the ability of dietary fibers to form carbohydrate structures that obstruct the interaction between intestinal pathogens and colonic epithelial cells [169]. Additionally, the microbial fermentation of dietary fibers produces SCFAs, such as acetate, propionate, and butyrate, which are crucial for maintaining colonic epithelial cell integrity and regulating the production of various cytokines by colonic epithelial cells [170].

However, the interactions between the diet and gut microbiota are complex and can sometimes lead to inflammatory responses [67]. For instance, highly fermentable fibers may induce pro-inflammatory effects, especially when provided at high doses [165]. The microbial-derived SCFA, butyrate, may exert regulatory effects on the composition of gut bacteria, leading to altered inflammatory responses [171]. Moreover, diets high in fat or sugar have been associated with increased gut permeability, dysbiosis, and chronic inflammation in animal models [172]. Similarly, additives or emulsifiers found in processed foods may disrupt gut physiology and promote inflammation by altering the mucosal layers that protect gastrointestinal epithelial cells [165].

Furthermore, dietary components can affect gut physiology, such as bile acid secretion, which has antimicrobial properties [60]. Certain food additives and minerals, like calcium phosphate, can also influence gut microbiota composition and homeostasis [173]. These interactions are particularly relevant in conditions like IBD, where disruptions in the mucosal and mucus layers exacerbate disease activity and chronic inflammation [174]. 

Therefore, understanding the intricate interplay among diet, gut microbiota, and inflammation is crucial for managing inflammatory and gastrointestinal diseases effectively.

## 7. Gut Microbiota Metabolites with Immunoregulatory Properties

SCFAs are organic acids with two to four carbon chains produced through the fermentation of dietary fiber by gut microbiota [163]. Among these, acetate, propionate, and butyrate are the most abundant, typically found in the colon at concentrations ranging from 10 to 100 mmol/L [153]. SCFAs play crucial roles in maintaining colonic epithelial homeostasis by serving as a primary energy source for colonocytes [175]. Moreover, they directly regulate the production of inflammatory cytokines by colonic epithelial cells [176].

Additionally, SCFAs have the ability to modulate the activity of various immune cells within the colonic mucosa, including macrophages, neutrophils, dendritic cells, T cells, and B cells [177]. This modulation can occur through several mechanisms, such as the engagement of SCFA-specific G-protein coupled receptors on the cell surface or intracellularly by modulating epigenetic regulation through the inhibition of histone deacetylases (HDAC) [178]. For instance, butyrate can inhibit HDAC in human cell lines at concentrations ranging from 10 to 100 µmol/L, while propionate and acetate exhibit similar properties at concentrations of 100 to 1000 µmol/L [179]. The inhibition of HDAC by SCFAs leads to the altered expression of transcriptional regulators of immune function, depending on the concentration of SCFAs present and the surrounding immune milieu [180].

Moreover, SCFAs are metabolized intracellularly to form metabolic intermediates such as acetyl-CoA, which drive metabolic and epigenetic changes upon immune cell activation [176]. These immune effects of SCFAs can also be observed systemically, as they can be absorbed into the peripheral circulation via portal veins within the gut, reaching serum concentrations ranging from 2 to 500 µmol/L [176]. Indeed, potent systemic anti-inflammatory effects have been observed in mice models of inflammatory disease and allergy when SCFAs are administered [165]. These effects are attributed to the downregulation of pathogenic responses driven by various immune cell subsets, such as B cells, Th1, Th2, and Th17, along with the concurrent upregulation of regulatory T cell (Treg) responses [181].

Furthermore, butyrate has been reported to promote regulatory B cells and inhibit the formation of germinal centers and plasmablasts, thereby reducing the severity of antigen-induced arthritis in mice [182]. By stopping the generation of proinflammatory molecules such as nitric oxide, IL-1b, TNFα, and nuclear factor kappa B (NF-κB) activity, butyrate causes human monocytes to produce more IL-10 and less IL-12 [183]. Additionally, butyrate activates caspase-8 and -9, which in turn causes neutrophil death and inhibits high mobility group box-1 [184]. Butyrate can influence various aspects of anticancer activity, including telomerase activation, GPR109a activation, and expression of the butyrate transporter sodium-coupled monocarboxylate transporter-1 (SMCT-1). In these ways, it can raise cancer cell apoptosis through a histone deacetylase (HDAC) inhibitor-dependent or independent pathway [185]. 

These findings underscore the significant immunoregulatory properties of SCFAs and their potential therapeutic implications in inflammatory diseases [186].

## 8. Protein Catabolite Breakdown of Dietary Protein by the Gut Microbiota 

Protein catabolism by the gut microbiota yields various metabolites, including branched-chain fatty acids (BCFAs), ammonia, phenols, and hydrogen sulfide [187]. Increased consumption of dietary protein, especially from animal sources, can lead to elevated levels of these catabolites [165]. Key microbial species involved in amino acid breakdown include *Escherichia*, *Eubacterium*, *Clostridium*, and *Enterococcus* [60]. Dysbiosis, characterized by an imbalance in microbial composition, may result in opportunistic pathogens like *Clostridium perfringens* and *Escherichia coli*. Studies on mice fed high-protein diets have shown reduced colonic immunoglobulin G (IgG) levels and an increased presence of *Escherichia coli* in the gut [29].

Exposure of colonic epithelial cells to ammonia and hydrogen sulfide can impair growth, increase cell turnover, and upregulate pro-inflammatory cytokine production, leading to enhanced epithelial permeability [165]. Hydrogen sulfide further amplifies T lymphocyte activation and interleukin-2 (IL-2) production upon T cell receptor (TCR) stimulation [188]. While hydrogen sulfide treatment exacerbates inflammation in septic mice, it paradoxically aids in resolving colonic inflammation by promoting the expansion of FOXP3+ regulatory T cells (Tregs) through the upregulation of methylcytosine dioxygenases Tet1 and Tet2 [189]. Moreover, secondary metabolism of BCFAs by microbes such as *Bacteroides fragilis* can yield lipid α-galactosylceramides, which regulate the function and abundance of CD1d-restricted natural killer T cells in murine gut tissue [190].

Tryptophan metabolism by the gut microbiota generates indole and its derivatives, which serve as ligands for the aryl hydrocarbon receptor (AhR) found in various immune cells [191]. Activation of AhR leads to the production of interleukin-22 (IL-22) while downregulating IL-17, thus promoting mucosal immune tolerance and epithelial integrity maintenance [192]. Patients with conditions like Crohn’s disease and coeliac disease may exhibit impaired tryptophan metabolism, resulting in defective AhR activation [192].

Furthermore, microbial degradation of immunogenic proteins such as gluten and amylase trypsin inhibitors can influence the T cell and innate immune response in coeliac disease [193]. While peptides produced by certain microbial species like *Pseudomonas aeruginosa* may exacerbate inflammation, degradation by others like *Lactobacillus* species can yield peptides with reduced immunogenicity, potentially offering protective effects against inflammation in coeliac disease [194].

## 9. Dietary Fiber, Gut Microbiota, and CRC

IBD, notably, ulcerative colitis, is a known precursor to CRC, which ranks as the third most common cancer globally [195]. CRC is influenced by dietary patterns, smoking, physical activity, and genetic and environmental factors [196]. Humans and their gut microbiota maintain a mutualistic relationship, yet this symbiosis can become pathological in conditions such as obesity, diabetes, atherosclerosis, IBD, and cancer [197]. CRC is believed to be associated with localized inflammation, which, while not capable of initiating oncogenesis independently, is considered a significant contributor to the process [198]. 

Studies have demonstrated that gut inflammation and the development of CRC can be influenced by dietary factors, gut microbiota composition, and the intestinal environment, presenting potential modifiable factors for altering CRC outcomes [199]. As alluded to before, SCFAs are beneficial metabolites produced by the local microbiota during the fermentation of fiber and other indigestible starches as they transit from the small intestine to the large colon [179]. These SCFAs possess anti-neoplastic, anti-inflammatory, and physiological properties, contributing to their potential role in CRC prevention and management [200].

Contrasting dietary habits among populations, such as higher animal protein and fat consumption among Americans versus increased carbohydrate and fiber intake among Africans, are reflected in health outcomes. African Americans exhibit higher rates of polyps and mucosal proliferation in the colon, potentially indicating elevated cancer risk [201]. These dietary disparities correspond to significant alterations in microbiota composition, with Americans showing dominance of the *Bacteroides genus* and Africans the *Prevotella genus*, along with distinct metabolic phenotypes [201]. Africans exhibit higher levels of starch-degrading bacteria, carbohydrate fermenters, and butyrate producers, whereas Americans harbor potentially pathogenic proteobacteria and bile acid deconjugators [201]. Plant-based diets rich in vegetables, fruits, cereals, nuts, and legumes, with moderate use of olive oil, fish, seafood, and dairy, and limited meat and alcohol intake, as observed in the Mediterranean diet, are associated with health benefits [202]. Increased consumption of high-fiber foods or fiber supplements correlates with lower blood pressure, improved blood glucose levels, weight loss, and reduced CRC risk [203] (Table 4). 

Fiber fermentation by gut bacteria yields beneficial metabolites, particularly butyrate, known for its anti-inflammatory and anti-neoplastic properties [161]. Conversely, secondary bile acids, produced from bacterial bile acid conjugation, possess carcinogenic potential [204]. Dietary fibers bind conjugated primary bile acids, potentially modulating gut bacteria involved in bile acid metabolism. This structural interaction may influence host physiology by preventing toxic bile acid accumulation, which can contribute to polyp formation and CRC, or by promoting bile acid destruction, triggering G protein-coupled receptor 5 (TGR5) and glucagon-like peptide-1 (GLP-1) production [205]. 

Chronic activation of GLP-1 receptor signaling has been linked to colon cancer development in animal models [206]. High-fiber foods, nuts, avocados, and eggs, rich in monosaccharides, peptides, amino acids, monounsaturated and polyunsaturated fatty acids, and short-chain fatty acids, may stimulate GLP-1 secretion, potentially exerting beneficial effects in healthy individuals [207]. Additionally, bacterial digestion of dietary fibers releases minerals and phenolic compounds that can be absorbed in the distal intestine [154].

Decreased consumption of dietary fiber diminishes microbial diversity and reduces the generation of SCFAs, shifting gut microbial metabolism towards less favorable substrates such as dietary and endogenous proteins [208]. This alteration may foster the proliferation of mucus-degrading bacteria, potentially compromising host health [209]. Studies have shown a marked decrease in total SCFA and butyrate production in human volunteers fed high-protein, low-carbohydrate diets [210]. Conversely, the fermentation of amino acids yields potentially harmful metabolites including branched-chain fatty acids, ammonia, N-nitroso compounds, p-cresol, sulfides, indoles, and hydrogen sulfide [211] (Table 4). These cytotoxic and pro-inflammatory metabolites contribute to the pathogenesis of chronic diseases, particularly CRC. CRC patients exhibit lower levels of butyrate-producing bacteria compared with healthy individuals, indicating a significant dysbiosis characterized by reduced butyrate production and the increased presence of pathogenic organisms [212].

**Table 4 ijms-25-08250-t004:** Dietary fiber, gut microbiota, and CRC.

Dietary Fiber	Effects	References
	1. Soluble fiber metabolism produces SCFAs, which have physiological, anti-inflammatory, and anti-neoplastic qualities that may help prevent and treat CRC.	[200]
	2. Beneficial metabolites, like butyrate, which has anti-inflammatory and anti-neoplastic effects, are produced when gut bacteria ferment fiber.	[201]
	3. Dietary fiber can be rich in polyunsaturated fatty acids, SCFAs, monosaccharides, peptides, and amino acids; high-fiber meals, nuts, and avocados may promote GLP-1 production, which may exert positive effects on healthy people.	[205]
	4. Numerous health benefits of dietary fiber include laxation, improved absorption of minerals, reduced inflammation, and potential protection against cancer.	[154]
	5. Dietary fiber also enhances viscosity and fecal bulking and reduces proteolytic fermentation time, limiting the interaction between potential carcinogens and mucosal cells.	[210]
	6. In the colon, dietary fiber can reduce the pH of feces and bind luminal carcinogens such secondary bile acids.	[211]
	7. In addition to increasing butyrate levels, dietary sodium gluconate reduces colon cancer rates.	[212]

SCFAs, short-chain fatty acids; CRC, colorectal cancer; GLP-1, glucagon-like peptide-1.

## 10. Mechanism Associated with Dietary Fiber Anticarcinogenic Activity

Increased carbohydrate intake promotes bacterial proliferation, resulting in faster colon transit times and decreased accumulation of potentially pathogenic chemicals such as ammonia, phenols, amines, and hydrogen sulfide in the colon [213]. Dietary fiber confers numerous health benefits, including laxation, mineral absorption, anti-inflammatory effects, and anticancer properties [214]. These advantages are largely attributed to the chemical degradation of dietary fiber into SCFAs within the colon [215]. Dietary fiber also enhances viscosity and fecal bulking and reduces proteolytic fermentation time, limiting the interaction between potential carcinogens and mucosal cells [216]. Additionally, dietary fiber may bind luminal carcinogens, such as secondary bile acids, and lower fecal pH in the colon [217]. Furthermore, dietary fiber serves as a source of vitamins, minerals, carbohydrates, and phytochemicals, which protect the gastrointestinal tract from oxidative stress [218].

Additionally, SCFAs, particularly acetate, propionate, and butyrate, act as ligands that bind to specific G-protein coupled receptors (GPCRs) on colonocytes and immune cells [219]. These SCFAs function as signaling molecules in the large intestine, reducing the production of proinflammatory cytokines and increasing the total number of regulatory T cells [170]. GPCR43 (FFAR2), GPCR41 (FFAR3), and GPCR109A are the primary GPCRs responsible for mediating the anticarcinogenic effects of SCFAs [220].

The molar proportion of SCFA components is influenced by dietary intake and the gut microbiota [221]. Acetate, primarily produced by a variety of bacteria in the colon, accounts for the majority of total SCFAs [222]. Propionate and butyrate, produced by specific bacterial groups, possess significant health benefits. These SCFAs exhibit anti-inflammatory properties and modulate various cellular processes associated with carcinogenesis [186].

Butyrate restricts HDACs, allowing for histone hyperacetylation, which results in the transcription of a number of genes, including p21/Cip1 and cyclin D3 [223]. Butyrate inhibits the migration and invasion of cancer cells by raising antimetastasis gene expression and reducing the activation of pro-metastatic genes at 0.5 or higher mmol/L concentrations [224]. Dietary fiber helps to prevent colon cancer in its early stages. Carbohydrates may protect colonocytes against genotoxicity caused by high-protein, high-fat Western diets. As a result, resistant starch reduces DNA damage in colonocytes expressed by single-strand breaks by 70% [217]. If this DNA damage is not repaired, colonic carcinogenesis can occur, and resistant starch protects against it [225]. The increased generation of SCFAs, as well as lower phenol and ammonia levels, may explain the preventive role of resistant starch against such DNA changes [225]. Butyrate, one of the SCFAs, has been shown to have a biological effect on neoplastic colonic cells [226]. Dietary sodium gluconate raises the level of butyrate and lowers the number of colon cancers [227]. The oral administration of bacteria that produce butyrate Butyrivibrio fibrisolvens increased the level of butyrate in the colon and rectum and decreased the production of aberrant crypt foci, an early colonic lesion [228].

In conclusion, advancements in microbiome and immunological profiling offer prospects for more precise patient stratification, identifying cohorts poised to derive maximal benefit from dietary interventions. Ultimately, collaborative efforts among immunologists, clinicians, nutritionists, and dietitians are imperative for designing well-structured clinical trials to corroborate the clinical efficacy of dietary therapy in managing inflammatory diseases.

In summary, dietary interventions rich in dietary fibers may play a crucial role in cancer prevention, especially in CRC. Numerous studies underscore the pivotal role of dietary fiber consumption in overall metabolic health, acting through fundamental pathways like T regulatory (Treg) cells, the Wnt signaling pathway, and G protein-coupled receptors (GPCRs). Herein, we provide clear associations between dietary fiber intake and improved colon function, gut motility, and reduced CRC risk. The gut microbiota emerges as a critical mediator of the beneficial effects of dietary fiber, influencing aspects such as metabolic processes and chronic inflammatory pathways.

The deficiency of dietary fiber in the typical Western diet stems from several factors. A significant portion of the population has adapted to modern lifestyles characterized by diets rich in ultra-processed foods. However, our gut microbiota, much like our bodies, has not evolved to accommodate this nutritional shift. Our dietary choices serve as predictors of overall health and well-being, with various benefits mediated by our gut bacteria.

Notable concerns include the excessive consumption of carbohydrates and fats, as well as the glaring lack of dietary fiber in modern diets. As consumers, prioritizing high-fiber foods over fiber-poor ultra-processed options is poised to have a substantial positive impact on future health outcomes. This shift in consumer preferences is likely to influence the strategic commercial plans of food companies, leading to improvements in the fiber content of processed foods.

## 11. The Effect of Antioxidants on the Gut Microbiome

Humans’ adequate diets and lifestyles impact their general health [229]. The macro- and micronutrients in an adequate diet supply the body with the nourishment it needs to continue growing and developing. Macronutrients such as carbohydrates, fats, and proteins are responsible for supplying the human body with energy and structural strength [230,231]. Micro-nutrients (vitamins, minerals, and bioactive compounds) play crucial roles in modulating different biochemical, metabolic, and signaling pathways. Most micronutrients function as antioxidant compounds to fight against nitric and oxidative stresses [232]. The prevention of diseases such as cancer, diabetes, and cardiovascular diseases is tightly associated with the consumption of antioxidants in the diet [233,234]. 

The antioxidant compounds consumed by humans include polyphenols, vitamins, selenium, and zinc. Their inadequate consumption has many harmful effects on immune function because of the link with the appearance of different diseases [235,236,237,238,239]. Knowing the variety of the gut microbiota and understanding its role in the modulation of specific pathological human states has become essential and can offer new strategies for developing novel therapeutic approaches [29]. The microbiota species, their specific effects, and the constant equilibrium among the different species can significantly affect the functionality of the body’s immunity and finally impact the human state of health [240]. Most bacteria are divided into three phyla including *Actinobacteria*, *Firmicutes*, and *Bacteridetes*. They represent 98% of all microbiota [241]. 

An individual’s gut microbiome usually consists of a few hundred microbial species depending on the characteristics of bacterial species, strain-level diversities, and numbers, although there is homology in the various human intestinal microbial genes [25]. In this particular case, it has been demonstrated that antioxidant compounds can protect against reactive oxygen species (ROS) and affect the microbiota arrangement of species. Because of these changes, human health is significantly improving [242]. Different enzymes that are responsible for hydrolyzing glucuronides, esters, glycosides, sulfates, lactones, and amides are found in the intestinal microbiota [243]. 

Food complexes include different molecules broken down further by bacteria in the gut to extract essential vitamins and amino acids. These nutrients are involved in many physiological activities of the host, including immunity and host energy metabolism [60]. Many studies have shown that ROS cause microbial dysbiosis, which can also lead to several intestinal disorders, such as intestinal mucosa damage, CRC, IBD, and enteric infections [244]. Testing the microbial molecular phylogeny (16S rRNA) allows for the detection of different classes of gut microorganisms including *Bacteria*, *Archaea*, and *Eukarya*; nonetheless, the majority of the human gut microorganisms in the system are bacteria. It has been suggested that the microbiota of the majority of individuals can be classified into one of three variants, or “enterotypes,” based on the dominant genera (*Bacteroides*, *Prevotella*, or *Ruminococcus*), despite the wide variability in the taxa present in the gut and interindividual variability in microbial composition [245]. 

Most bacteria in the human gut belong to the phyla *Bacteroidetes*, *Firmicutes*, *Proteobacteria*, and *Actinobacteria*. Over 90% of the bacteria found in the human gut belong to the taxa *Bacteroidetes* and *Firmicutes*. Furthermore, most of the gut bacteria in the *Archaea* domain are *Methanobrevibacter smithii*. Individual differences exist in the diversity of gut microorganisms because factors including dietary habits, living conditions, and physical well-being mostly cause these differences [152]. 

Vegetables are rich in compounds with antioxidant characteristics, which may reduce the damaging impacts of the redox processes that every cell goes through as a result of its biochemical responses. This is especially true if the vegetables are not subjected to chemical or technological treatments. Consequently, a greater quantity of these nutrients can be found in vegetable-based diets like the Mediterranean diet than in modern ones like the so-called “cafeteria diet” or the typical Western diet that most people follow [246]. According to Lobo et al. [247], antioxidants are high electron density compounds that have the ability to neutralize free radicals by allowing them to give up one electron in order to quench the free radical. 

Researchers have indicated that antioxidants, or free radical inhibitors, participate with free radical molecules in a series of chain reactions that result in the complex formation of blocker and free agent molecules that generate stable molecules [248]. A variety of enzymes, including glutathione peroxidase, catalase, and superoxide dismutase, are found in the cells of the biological system and are capable of neutralizing free radicals [249,250]. Furthermore, excessive consumption of animal-derived food may increase the risk of obesity and its related non-communicable diseases because of microbial alterations and the production of inflammatory metabolites (TMAO) [251,252]. Additives from processed food can affect the liver, nervous system, microbiota, and intestines because of oxidative stress [253]. Low-calorie density and a high content of micronutrients are two characteristics of fruits and veggies [254]. 

Obesity and metabolic dysfunction are closely linked to heightened ROS production within the gastrointestinal (GI) tract, exacerbating dysbiosis in the gut microbiota. Recent research has shown that dysbiosis can be reversed through dietary supplementation with antioxidants, particularly polyphenols. Polyphenols are believed to mitigate GI ROS levels, suggesting radical scavenging within the GI tract as a potential mechanism for correcting dysbiosis. The study conducted by Van Buiten et al. [255] aimed to explore the relationship between GI ROS, dietary antioxidants, and the beneficial gut bacterium *Akkermansia muciniphila*. Their results indicated that *A. muciniphila* discriminates between lean and obese mice. Additionally, the relative abundance of A. muciniphila displayed a significant negative correlation with extracellular ROS levels in the GI tract. Their study evaluated the abilities of dietary antioxidants—ascorbic acid, β-carotene, and grape polyphenols—to scavenge GI ROS and promote *A. muciniphila* blooms in lean mice. While the inverse relationship between GI ROS and *A. muciniphila* abundance held true in lean mice, only grape polyphenols were effective in stimulating *A. muciniphila* proliferation. Analysis of fecal antioxidant capacity and antioxidant bioavailability suggested that the superior radical scavenging activity and support for *A. muciniphila* by grape polyphenols may be attributed to their comparatively limited bioavailability. These findings underscore the potential of modulating the GI redox environment as a therapeutic strategy for chronic inflammatory conditions such as metabolic syndrome.

In conclusion, the state of total health is reflected in the reciprocal and bidirectional relationships among food, dietary antioxidants, inflammation, and body composition (obesity).

## 12. Dietary Antioxidants—Polyphenols

The interaction among polyphenols and many metabolic pathways led to the generation of metabolites that have been demonstrated to impact the gut microbiota positively by encouraging the growth of beneficial bacteria. Different metabolic pathways associated with the prevention of type 2 diabetes and cardiovascular disease are affected by polyphenols [256]. Polyphenols can be converted by gut microorganisms into bioactive compounds that affect the intestinal ecology [257]. A recent research study described two synergistic actions of polyphenols responsible for suppressing detrimental gut bacteria and concomitantly supporting the growth of favorable bacteria responsible for boosting the absorption of polyphenols. 

Polyphenols not only block many dangerous bacterial species like *Clostridium* and certain pathogenic species like *Helicobacter* and *Salmonella*, but they also stimulate the growth of helpful *Lactobacillus* species [258,259]. Additionally, according to the authors, polyphenols preserve gut health by controlling the makeup of microbes. Usually directly absorbed in the small intestine, they have an impact, or the gut bacteria converts the unabsorbed polyphenols, enhancing host health [260]. Gram-positive bacteria are more likely to proliferate during gut microbial colonization when polyphenols are present as opposed to Gram-negative bacteria [261]. Secondary plant metabolites known as polyphenols are found in large quantities in a variety of foods, including tea, coffee, wine, cocoa, spices, and a variety of vegetables and fruits. 

Fruits and vegetables have the greatest polyphenol concentration of all those sources. In most regions, the average daily consumption of polyphenols is 900 mg/kg/body weight [262]. According to Sorrenti et al. [263], only 5–10% of polyphenols are absorbed in the small intestine; the remainder are broken down by gut bacteria in the colon [263]. Although polyphenols are well known for their antioxidant qualities, there has been much research over the past ten years on how they interact with gut bacteria [264]. Polyphenols are classified into the following three groups based on their chemical structure: flavonoids, phenolic acids, and non-flavonoids [264]. Most polyphenols are found as glycones and aglycones, with a small amount also existing as condensed tannins [265]. 

Benzoic acid and cinnamic acid groups differentiate phenolic acids, which make up the majority of polyphenols [263]. Six subclasses comprise the flavonoid group including the following: flavanols, flavones, isoflavones, flavonols, flavanones, and anthocyanidins. In comparison with low molecular weight polyphenols, larger molecular weight polyphenols are absorbed more slowly [266]. In a similar vein, colonic bacteria break down the aromatic ring of phenolic acid, yielding absorbable aglycones [267]. Furthermore, Catalkaya et al. [268] reported that the multienzymatic reactions of gut microbes result in catabolic transformations, such as the C-C breakage of the aromatic ring of polyphenolic compounds [268]. 

Epicatechin and 4-hydroxybenzoic acids, two molecules of lower molecular weight, were produced when complex polyphenols were broken down by the gut [269]. It was reported that the flavonoids coupled to sugar moieties need to enter the colon, where they are hydrolyzed for improved absorption by colon microbial enzymes. According to the authors of that study, enzyme esterases are necessary to degrade polyphenols esterified with sugar moieties and other components. Since humans are not able to metabolize esterases, colon microbial esterases are mostly in charge of breaking down ester connections [270]. Analogously, gut microorganisms produce enzymes that aid in absorption, such as lactase, phlorizin hydrolase, and β-glucosidase, which hydrolyze glycosylated flavonoids. *Enterococcus*, *Streptococcus*, *Lactobacillus*, *E. Coli*, and *Bifidobacterium* are some bacterial species in gut colonization linked to polyphenol metabolism [259]. In the intestinal segment of the GI tract, where intestinal microbial enzymes also play a crucial part in the uptake of polyphenols, a smaller quantity of polyphenols are assimilated. The growth of *Lactobacillus* and other beneficial bacteria increased while pathogenic bacteria like *E. coli*, *S. typhimurium*, and *L. monocytogenes* decreased in Peng et al.’s investigation of the impact of cocoa polyphenols on probiotic bacterial growth [271]. Additionally, *Salmonella*, *Clostridium*, *Bacillus*, and *Streptococcus* were among the bacterial species against which epigallocatechin gallate (EGCG) showed an inhibitory effect in the gut [272]. 

In a different study, Murota et al. found that dihydroxylphenylacetic acid (DOPAC), a metabolite of the powerful neurotransmitter dopamine, is produced when intestinal fecal bacteria like *Clostridium* and *Bacteroids* break down quercetins [273]. A decrease in bacterial species like *Bacteroides*, *Parabacteroides*, and *Alistipes* was noted by the authors of an in vivo investigation of the impact of tart polyphenol consumption on gut microbiota [269]. According to Moorthy et al., the breakdown product of ellagitannins by the gut microbiota, urolithin, is formed when pomegranate juice extract is consumed through the diet [274]. This leads to a considerable increase in the composition of *Gordonibacter* in the intestinal tract. A different investigation by Selma et al. found that the dibenzopyranone metabolite urolithin had anti-inflammatory properties in vivo [275]. 

Conversely, an investigation used a double-blind, placebo-controlled trial to examine the composition of the gut microbiota using a 16S rRNA sequencing technique following Aronia fruit consumption. It was demonstrated that the polyphenols in Aronia berries regulated the gut microbiota by substantially stimulating the proliferation of *Anaerostipes* and *Bacteroides*. Gram-positive bacteria discovered in the gut called anaerostipes have been shown to defend against colon cancer [276]. 

The human gut microbiota, which includes bacteria comprising *Bacteroides*, *Lactobacillus*, *Enterococcus*, *Bifidobacterium*, and *E. coli*, was exposed to various concentrations of polyphenols, and Duda-Chodak calculated their minimal inhibitor concentrations (MICs). A comparison of quercetin with other flavonoids, including naringenin, hesperidin, rutin, and catechin, revealed that the former had the greatest inhibitory potential against human gut flora [277]. The consumption of wine or grape polyphenols was found to lower the *Firmicutes/Bacteroidetes* ratio in the in vivo investigation [278]. The amount of *Bacteroidetes* was also raised by the ingestion of 60 mg/kg of chlorogenic acid [279]. Extremely high concentrations of polyphenols in olive oil showed an inhibitory impact on *Helicobacter pylori* in another in vitro study on the gut bacteria [280]. *L. monocytogenes* and *S. enteritidis* were reduced in their growth by an extract from blueberries high in quercetin and chlorogenic acid (112.5–900 mg/mL) [281]. 

In summary, based on the aforementioned experimental findings and additional published data, it was found that dietary polyphenol intake affected a small number of bacterial species, including *Bifidobacteria* and *Lactobacillus*, which are thought to be beneficial to the gut health of humans. In this regard, numerous polyphenols have demonstrated an inhibitory effect on the proliferation of pathogenic bacterial species, including *Bacteroides* and *Clostridium* [282]. 

### 12.1. Polyphenols and Inflammation in Obesity 

Numerous investigations, both in vitro and in vivo, have explored the anti-inflammatory characteristics of polyphenols. These investigations have indicated that polyphenols can target several enzymes, such as phospholipase A2, lipoxygenases, and cyclo-oxygenases (COXs), which in turn reduce the synthesis of prostanoids and leukotrienes [283]. Furthermore, transcriptases, kinases, and phosphodiesterases are additional focuses of polyphenols’ anti-inflammatory action. Studies have indicated polyphenols’ ability to modify the inflammatory status associated with obesity in addition to their general anti-inflammatory properties. 

Obesity is without a doubt the primary metabolic condition that is most closely linked to persistent low-grade inflammation [284,285]. Increased circulating levels of inflammatory biomarkers, mostly generated by monocytes, macrophages, and defective adipocytes, are evidence of this in obese individuals compared with non-obese participants [286,287]. This is due to the stimulation of pro-inflammatory genes’ molecular expression by NF-κB. To be more precise, NF-κB is a family of transcription factors that is generally found in the cytoplasm and is linked to IκB, a family of regulatory proteins that includes Bcl-3, IκBγ, IκBβ, IκBα, and IκBε. It also includes NF-κB1 and NF-κB2 [288,289]. NF-κB is kept inactive when it is linked to IκB. Certain genes, including NF-κB-inducing kinase (NIK), mitogen-activated protein kinase kinase (MEKK), interleukin-1 receptor-associated kinase (IRAK), TNF receptor-associated factor (TRAF), PKC, and VCAM, promote the activation of IKK, which phosphorylates IκB and releases NF-κB in response to different stimuli (e.g., increasing oxidative stress and/or inflammation) [290]. Pro-inflammatory genes are expressed when NF-κB is produced and translocates into the nucleus [291]. Evidence has shown that polyphenols can function at different levels to affect the NF-κB pathway in this situation [292]. Specifically, mechanistic research has revealed that polyphenols can (i) prevent IKK from being activated (EGCG, epicatechin, and flavonoids) [293]; (ii) prevent IκB from being phosphorylated (EGCG, quercetin, apigenin, silymarin, kaempferol and isoliquiritigenin, curcumin) [294,295,296]; (iii) prevent IκB from being degraded (EGCG, epicatechin, quercetin, apigenin, and isoliquiritigenin) [293,297]; (iv) prevent NFκB from being nuclear translocation (quercetin and isoliquiritigenin) [283,297]; and (v) prevent NFκB from binding DNA (EGCG, epicatechin, quercetin, and apigenin) [283,298,299]. One intriguing mechanism that is implicated in the inflammatory response is the “mitogen-activated protein kinase (MAPK) cascade” [300,301]. Ser/Thr kinases include proteins like MAPKs, which modulate gene expression in response to particular stimuli to regulate a variety of cellular activities [292]. It has been documented that polyphenols act at several levels to modulate the MAPK pathway [302]. Specifically, it has been demonstrated that many MAPKs in the MAPK cascade are inhibited by polyphenols such as EGCG, kaempferol, chrysin, apigenin, luteolin, quercetin, catechin, and cyanidin-3-O-glucoside [283]. These polyphenols reduce the transcription of pro-inflammatory cytokines. 

Resveratrol was found to have a strong anti-inflammatory impact in two intriguing meta-analyses [303,304]. Remarkably, neither of the two meta-analyses found any evidence that resveratrol significantly lowers IL-6 levels [303,304]. Only a significant decrease in TNFα and hs-CRP was observed after the resveratrol treatment. 

It can therefore be concluded that different polyphenols exert a strong anti-inflammatory effect, particularly in obesity. Obesity is directly associated with inflammation followed by enhanced carcinogenesis [305].

### 12.2. Polyphenols and Inflammation in Cancer 

According to predictions of the World Health Organization, 10 million people died from cancer in 2020 [306]. Among the most common cancer types were skin (non-melanoma) (1.2 million cases), stomach (1.2 million cases), lungs (2.21 million cases), colon/rectum (1.93 million cases), prostate (1.4 million cases), and breast (2.26 million cases) [306]. It is noteworthy that Middle Eastern countries are witnessing a concerning increase in the cancer incidence rate. It has been projected that the cancer rate will double by 2030 [307,308]. However, compared with Western countries like the U.S., the incidence of occurrence is much lower [308,309]. 

We know significantly more about these beneficial compounds now than we did in the initial research period on the anticancer effects of various polyphenols [310,311]. These compounds are therapeutic and fascinating since they target many disease hallmarks and destroy cancer cells via different pathways. Berries, grapes, olive oil, cocoa, nuts, peanuts, and other fruits and vegetables are the primary providers of polyphenols, with a content between 200 and 300 mg/100 g fresh weight. Additionally, these fruits are used to prepare different foods, including wine, beer, and tea, all of which have high polyphenol content. The specific biological, chemical, and physical properties of each class of polyphenols are determined by the quantity and qualities of phenolic groups [312].

Cancer development has the following three main phases: initiation, promotion, and progression [310,311]. A carcinogenic chemical is consumed or exposed to interact with chromatin to create a mutation or epigenetic modification. This is the typical and quick first stage of the genesis of cancer [310,311] (Figure 3). 

Progress in cancer therapy has been limited by drug resistance, high treatment costs, and an increasing number of reports on additional toxicity, despite efforts to increase awareness of carcinoma, stimulate early detection, and create new effective therapies [313,314]. Further driving up treatment costs is the fact that the majority of chemotherapeutic drugs have a history of unpleasant side effects, including but not limited to nausea, vomiting, headaches, musculoskeletal pain, anorexia, gastritis, oral ulcers, diarrhea, constipation, alopecia, and neuropathy [315,316]. Antioxidants are compounds that help a target cell prevent, delay, or eliminate oxidative damage. 

In summary, antioxidants’ overall mode of action includes (a) reducing the generation of ROS via chelating trace elements or blocking the enzymes that produce free radicals; (b) scavenging ROS; and (c) upregulating antioxidant enzymes that provide protection [317].

### 12.3. The Antioxidant Activity of Flavonoids

The antioxidant activity of flavonoids is responsible for their ability to protect biological organisms. Research has shown that flavonoids can avoid damage caused by free radicals via the following mechanisms: (i) Direct scavenging of ROS, where flavonoids can directly scavenge free radicals by donating hydrogen atoms [318]. (ii) The suppression of oxidases including xanthine oxidase, cyclooxygenase, lipoxygenase, microsomal monooxygenase, glutathione S-transferase, mitochondrial succinoxidase, and NADH oxidase, which produce superoxide anions [319,320]. (iii) Modulation of phase II detoxifying enzymes, such as NAD(P)H-quinone oxidoreductase, glutathione S transferase, and UDP-glucuronosyl transferase, which are the main defense enzymes against electrophilic toxicants and oxidative stress. Their activation involves the induction of these enzymes [321,322]. (iv) Metal traceability track. Regarding the metabolism of oxygen, this pathway is important. The reduction of hydrogen peroxide with the production of the extremely reactive hydroxyl radical and copper-mediated low-density lipoprotein (LDL) are two examples of how free iron and copper may increase the synthesis of ROS [323]. Some flavonoids can reduce the likelihood of free radical formation by chelating to iron and copper metals. (v) Nitric oxide-induced oxidative stress mitigation. Although nitric oxide (NO) plays a crucial role in preserving blood vessel dilatation [324], increased amounts of NO may cause oxidative damage. NO synthases (NOSs) mediate the oxidation of l-arginine, which produces NO. Peroxynitrite is generated when nitric oxide (NO) reacts with superoxide (O_2_^−^). It is the primary mediator of nitric oxide toxicity [325]. Apigenin, disometin, and luteolin are examples of flavonoids that have been shown to decrease NO synthesis in human lipopolysaccharide-activated cell lines and cultures.

Benzoic and cinnamic acids are **phenolic acids**. Although the exact biosynthetic pathway for benzoic acids is unknown, they are often generated from cinnamic acid and its byproducts. The most common examples are p-hydroxybenzoic acids, which are produced by the decomposition of tyrosine and include vanillic, protocatechuic, syringic, and gallic acid, as well as gentisic acid [326,327]. On the other hand, cinnamic acids are ortho-oxygenated and then methylated, which results in the formation of most hydroxycinnamic acids, including ferulic, p-coumaric, caffeic, and sinapic acid. Phenolic acids can exist in their free forms or conjugations with ethers, esters, and a range of other molecules, including plant polymers, simple sugars, and organic acids [328,329,330]. The presence of unsaturated substituted chains, the aromatic ring, and the quantity and orientation of free hydroxyl groups are important structural elements necessary for the anticancer properties of phenolic agents [326]. Since they are effective in treating a wide range of illnesses, including diabetes, cancer, and cardiovascular and neurological disorders, phenolics are well-liked as therapeutic substances. Through the modification of glucose metabolism, their antidiabetic effect is mediated [328]. Their strength as anticancer agents is mainly ascribed to their antioxidant activity; they are potent radical scavengers, metal chelators, alterers of endogenous defense mechanisms like superoxide dismutase (SOD), catalase (CAT), and glutathione peroxidases (GPx), enhancers of the redox status of glutathione (GSH), and modulators of various proteins and transcriptional factors like nuclear factor erythroid related factor (NRF2) [328,331]. Additionally, their capacity to suppress angiogenic factors (vascular endothelial growth factor; VEGF; MIC-1); oncogenic signaling cascades (phosphoinositide 3-kinase; PI3K; protein kinase B; Akt); induce apoptosis; and inhibit cell proliferation (extracellular signal-regulated kinase; Erk)1/2, D-type cyclins, and cyclin-dependent kinases (CDKs)) are linked to their anticarcinogenic properties [328,329] (Table 5). 

Vanillin is an intermediary molecule produced during the reaction that produces **vanillic acid** (4-hydroxy-3-methoxybenzoic acid, VA) from ferulic acid [332]. Green tea and *Angelica sinensis* are important sources of this significant active ingredient [332]. The antioxidant activity of VA, which eliminates free radicals, suggests that it functions as a chemo-protectant and can prevent lung cancer in Swiss albino mice caused by benzo(a)pyrene [333]. In Dimethyl-1,2, benzanthracene (DMBA)-induced hamster buccal pouch carcinogenesis, VA also showed antioxidant and anti-lipid peroxidative properties. It is interesting to note that after administering DMBA alone, changes in the levels of lipid peroxidation by-products and anomalies in antioxidative status were observed; however, they were restored after administering VA [334]. One study investigated whether whole plants or isolated plant extracts containing a variety of natural chemicals have any anticancer properties. The transformed root extract of *Leonurus sibiricus*, which is rich in gentisic acid, 4-hydroxybenzoic acid, 1,3-dicaffeoylquinic acid, and α-resorcylic acid, among other polyphenolic compounds, has been shown to be able to inhibit the proliferation of glioma cells and induce apoptosis. This is achieved through the generation of ROS, loss of mitochondrial membrane potential, enhancement of the S and G2/M cell cycle phases, and modifications in the mRNA levels of apoptotic factors, such as Bax, Bcl-2, p53, and caspase-3, -8, and -9 [335]. Isolated VA had a dose-dependent (30 µM) impact on the expression of hypoxia-inducing factor (HIF-1) protein synthesis, which prevented angiogenesis, proliferation, and cell cycle arrest at the G1 phase in HCT116 CRC cells. This effect was independent of HIF-1 protein degradation and mRNA expression. Therefore, the suppression of the Raf/MEK/ERK and mTOR/p70S6K/4E-BP1 pathways must occur for this inhibitory action [332].

A result of tyrosine catabolism, gentisic acid (2,5-dihydroxybenzoic acid, GeA) is a biosynthetic derivative of salicylic acid that can be found in citric fruits, grapes, artichokes, sesame, and olives. In addition to being used to treat cardiovascular disorders, it possesses anti-inflammatory and antioxidant properties [336,337]. GeA has strong antioxidant properties that may be attributed to its role as a free radical scavenger or to its agonistic interaction with NRF2, which controls the production of antioxidant compounds [336]. The extract of *Vaccinium myrtillus* shows anticancer activity on HCT-116 cells, which suggests that it is a potent antioxidant. It is noteworthy that, along with quercetin and kaempferol, GeA is thought to be one of the most prevalent polyphenols in this extract [338]. Additionally, in vitro growth, DNA synthesis, and colony formation in C6 gliomas were inhibited by treatment with GeA at clinically relevant levels. When taking into consideration the in vivo model, GeA improved the survival of Ehrlich breast ascites carcinoma-bearing mice. Regarding the development of Ehrlich solid tumors (ESTs) impacted by either GeA on its own or with sodium selenite, GeA inhibited the tumor growth while leaving the antineoplastic effects of selenium unaltered, which at first diminished EST. Nevertheless, at a later stage, GeA blocked the tumor stimulation caused by selenite. Finally, doxorubicin side effects such as myofibrillary and endothelial damage and hyalinization necrosis were decreased by GeA [339]. Furthermore, GeA indirectly regulates brain glioblastoma by inhibiting OAT3 and solute carrier-22A8, which are required for anticancer medication brain efflux. This accumulation of chemotherapeutic medicines in the brain slows down tumor growth. GeA and its isomers also control the course of the cell cycle by inhibiting the enzymatic activity of CDK 1 [336] (Table 5).

The chemical compound known as **gallic acid** (3,4,5-trihydroxy benzoic acid, GA) can be extracted from a variety of foods and beverages, including vinegar, wine, green tea, blackberries, raspberries, walnuts, chocolate, and chestnut-green chicory. It can also be found in free form [340]. Gallic acid has significant antibacterial, anti-inflammatory, and anticancer properties [341]. It primarily inhibits cellular proliferation, ROS production and propagation, and cell cycle arrest in the G2/M phase to achieve its anticancer effects [329,342]. By inducing apoptosis in MCF-7 breast cancer cells in the sub-G1 phase, an aqueous extract from *Rhus verniciflua* (RVSE) elevated p53 and p21. Notably, high-performance liquid chromatography of RVSE only detected the presence of GA; as a result, GA’s actions on MCF-7 cells may be responsible for their antiproliferative effects [343]. 

Syringic acid (4-hydroxy-3,5-dimethoxybenzoic acid, SyA) is an abundant phenolic compound present in dates, olives, pumpkins, grapes, spices, acai, red wine, palms, and honey [344]. It has higher potency than p-hydroxybenzoic acid because of the free radical scavenging activity attributed to the presence of two methoxy moieties at positions 3 and 5 [331]. Non-melanoma skin cancer is related to excessive UV exposure. A study focusing on an association between SA and UVB-induced signaling and skin cancer revealed that chemo-preventive potential in vitro and in vivo was mediated mainly via the ability of SA to inhibit the Nox/PTP-κ/EGFR axis [344]. The creation of iron–SA complexes, which block Fenton-induced oxidative damage by preventing the generation of free hydroxyl radicals, may be connected to the antioxidant activity of SA, which is an intriguing finding [345]. Additionally, by inhibiting the generation of ROS (superoxide dismutase) and reducing cell adhesion by reducing attachment to extracellular matrix (ECM) in human lung A549 and colon adenocarcinoma HT29-D4 cells, SA has been shown to have antioxidant properties [346]. 

**Table 5 ijms-25-08250-t005:** The antioxidant activity of flavonoids.

Compound	Effects	References
Benzoic and cinnamic acids	1. Strong antioxidants, metal chelators, modifiers of endogenous defense systems such as superoxide dismutase (SOD), catalase (CAT), and glutathione peroxidases (GPx), enhancers of glutathione’s redox status, and modulators of different proteins and transcription factors such as nuclear factor erythroid related factor (NRF2) are just a few of their many biological properties.	[328,331]
Vanillic acid (VA)	2. The expression of hypoxia-inducing factor (HIF-1) protein synthesis was affected by VA in a dose-dependent (30 µM) manner in HCT116 CRC cells, preventing angiogenesis, proliferation, and cell cycle arrest in the G1 phase.	[332]
Gentisic acid (GeA)	3. Since GeA scavenges free radicals and interacts agonistically with NRF2, which regulates the synthesis of antioxidant molecules, it possesses potent antioxidant qualities.	[336]
Gallic acid (GA)	4. GA’s anticancer effects are mediated mainly through the inhibition of cell cycle arrest in the G2/M phase, ROS formation and propagation, and cellular proliferation.	[329,342]
Syringic acid (SyA)	5. SyA has been demonstrated to possess antioxidant qualities by preventing the production of ROS (superoxide dismutase) and decreasing cell adhesion by lowering attachment to extracellular matrix (ECM) in human lung A549 and colon cancer HT29-D4 cells.	[346]

SOD, superoxide dismutase; CAT, catalase; GPx, glutathione peroxidases; NRF2, nuclear factor erythroid related factor 2; HIF-1, hypoxia-inducing factor; ROS, reactive oxygen species; ECM, extracellular matrix.

In conclusion, flavonoids’ capacity to protect organisms is due to their antioxidant activity. Research studies have indicated that flavonoids have several ways of preventing the harmful effects that ROS can have.

### 12.4. Non-Flavonoid Compounds and Cancer 

#### 12.4.1. Stilbenes 

Red wine and grapes are common sources of polyphenols, a non-flavonoid compound resveratrol, which has potent anti-aging and antioxidant qualities [347]. Multiple studies have shown that the co-administration of resveratrol with other therapeutic agents (paclitaxel, docetaxel, doxorubicin, rapamycin, and gefitinib) reduces multi-drug resistance (MDR) in colorectal, breast, and lung cancer. This occurs by improving the absorption of chemotherapeutic agents, prolonging drug retention, inducing pro-apoptotic mechanisms, arresting the cell cycle, and decreasing ABC transporters [348,349,350,351] (Table 6). 

#### 12.4.2. Ellagitannins 

Ellagitannins and their metabolite ellagic acid can inhibit P-gp, MRP, and BCRP proteins. In this way, they are responsible for the decrease in MDR in cancer [352]. Through an increased Bax/Bcl-2 ratio, caspase-3 activation, and the loss of mitochondrial potential, ellagic acid sensitizes human CRC cells to 5-FU therapy [353]. A crucial part of defeating MDR in malignant breast cell types is provided by ellatannins and their metabolites [354]. Agrimoniin, sanguiin-H6, tellimagrandin I, rugosins A and D, and pedunculagin are among the ellagitannins that Berdowska et al. investigated for their effects on doxorubicin-resistant breast cancer cells. Only sanguiin-H6 exhibited cytotoxic effects on resistant MCF-7 cancer cells out of all the compounds tested. This was likely because sanguisorbic acid dilactone was released, inhibiting ABC transporters and reducing the cells’ capacity to extrude other products of sanguiin-H6 hydrolysis, such as ellagic acid and depsides, which also had cytotoxic effects [354].

#### 12.4.3. Lignans

By downregulating P-gp expression and increasing membran fluidity, co-encapsulating paclitaxel and honokiol—a lignan extracted from the bark, stem, and leaf from *Magnolia* sp.—in pH-sensitive micelles made from polymers inhibited MDR in breast cancer [355]. Additionally, it increased the Bax/Bcl-2 ratio and levels of apoptosis (caspase-3 activation) and decreased expression of cyclin A1 and D1, which explains why honokiol radiosensitizes CRC [356]. The chemosensitivity of CRC cells to 5-FU was increased by other lignans, such as schizandrin A, which was extracted from Schisandra chinensis fruits and upregulated miR-195. Furthermore, NF-κB and PI3K/AKT signaling pathways were inactivated by miR-195 overexpression [357]. The primary active ingredient in milk thistle fruit silymarin, a blend of flavonolignans, is silybin. By inhibiting the important oncogenic pathways STAT3, AKT, and ERK, silybin therapy of breast cancer cells resistant to doxorubicin/paclitaxel sensitized cells to chemotherapeutic drugs [358]. Recent findings indicate that the cytotoxic effects of docetaxel, carboplatin, and doxorubicin in metastatic breast cancer cell lines were improved by the use of flaxseed lignan (secoisolariciresinol) and its metabolite (enterolactone), most probably by suppressing fatty acid synthase [359]. 

#### 12.4.4. Hydroxy-Cinnamic Acids

Reduced MRP1, P-gp, and BRCP expression in human colorectal malignancies is caused by ferulic acid and caffeic acid extracted from foxtail millet. Caffeic acid phenyl ester, or CAPE, is an effective suppressor of human breast tumor stem cells, preventing cell division and proliferation. Also, it can reduce the quantity of CD44+ cells. Tumor growth originates from a small number of CD44+ cells that are resistant to treatment [360]. CAPE increases the radiosensitivity of breast cancer cells, according to Khoram et al. [361]. Additionally, CAPE is effective in combating MDR in lung and prostate cancer by downregulating the synthesis of claudin-2, suppressing the NF-κB pathway, diminishing cytoplasmic reserves of GSH (reduced glutathione), and decreasing apoptotic regulators (cIAP1, cIAP-2, and XIAP) [362,363]. Recent studies have shown that treating lung adenocarcinoma-derived stem-like cells with cinnamic acid inhibits their proliferation and promotes their differentiation into CD133-negative cells, primarily isolated from carcinomas using this marker [364] (Table 6). 

**Table 6 ijms-25-08250-t006:** Non-flavonoid compounds and cancer.

Compounds	Effects	References
Stilbenes	1. Numerous studies have demonstrated that the co-administration of resveratrol with other therapeutic drugs, such as paclitaxel, docetaxel, doxorubicin, rapamycin, and gefitinib, can decrease multi-drug resistance (MDR) in colorectal, breast, and lung cancer.	[348,349,350,351]
Ellagitannins	2. Ellagic acid sensitizes human CRC cells to 5-FU therapy by increasing the Bax/Bcl-2 ratio, activating caspase-3, and reducing mitochondrial potential.	[353]
Lignans	3. For breast cancer cells resistant to doxorubicin/paclitaxel, silybin treatment sensitized the cells to chemotherapeutic medicines by blocking the key oncogenic pathways STAT3, AKT, and ERK.	[358]
Hydroxy-cinnamic acids	4. By suppressing the synthesis of claudin-2, suppressing the NF-κB pathway, reducing cytoplasmic reserves of GSH (reduced glutathione), and decreasing apoptotic regulators (cIAP1, cIAP-2, and XIAP), caffeinic acid phenyl ester (CAPE) is effective in fighting MDR in lung and prostate cancer.	[362,363]

MDR, multi-drug resistance; NF-κB, nuclear factor kappa-light-chain-enhancer of activated B cells; GSH, reduced glutathione; CAPE, caffeinic acid phenyl ester; STAT3, signal transducer and activator of transcription 3; AKT, protein kinase B; ERK, extracellular signal-regulated kinase.

Thus, because of their antioxidant properties, ability to inhibit angiogenesis, metastasis, proliferation, survival, and inflammation, ability to modulate immune and inflammatory responses, and capacity to inactivate pro-carcinogens, phenolic compounds (flavones, ellagitannins, stilbenes, lignans, etc.) exert a chemopreventive effect [365].

## 13. Dietary Antioxidants—Vitamins and Cancer

The expression of vitamin synthetic pathways to obtain B1, B2, B3, B5, B6, B7, B9, and B12 have been investigated in the genomes of 256 common gut bacteria, and the results show that 40–65% of the host gut bacteria are able to participate in the production of these vitamins [366]. Additionally, it was hypothesized that the gut microbiota exchanged vitamins with one another, a capacity that appears to be essential for bacterial species that are unable to synthesize certain vitamins. 

Vitamins can be created from scratch by the prototrophic bacteria that live in the large intestine [367]. Auxotrophs depend on prototrophic bacteria to deliver vitamins and absorb food components in the host’s small intestine. Moreover, the equilibrium of the host intestinal microbiota is greatly dependent on the cross-feeding of auxotrophic and prototrophic bacteria [368]. There has been increasing evidence that the vitamins obtained from the gut microbiota are related to several diseases and are crucial in the dysregulation of the connection between vitamins and bacteria [369]. For instance, germ-free mice with vitamin deficiencies had longer prothrombin times, bleeding, and a 100% death rate, however, none of the aforementioned abnormalities were observed in animals that were raised [370]. 

Vitamins are crucial for regulating the microbiota’s immunological response, which can affect the composition of the gut microbiota or result in dysbiosis [371]. It was proposed that vitamin-treated mice had higher *Citrobacter rodentium* abundances because of Th17 response impairment [372]. Moreover, *Bifidobacteria* abundance in vitamin-supplemented parents exhibited statistically negative changes, whereas *Bifidobacteria fragilis* abundance showed positive trends [373]. Retinoic acid-related orphan nuclear receptor, a transcription factor specific to a certain lineage, is stimulated in response to signals from IL-6, TGF-β, IL-21, and IL-23, which in turn affect the proliferation of Th17 cells. Th17 plays a crucial role in inflammation and gut immunological homeostasis by secreting cytokines such IL-22, IL-17 A, and IL-17 F [374]. Through a decrease in IL-17 production, vitamins may be able to prevent and even partially reverse experimental autoimmune uveitis. 

Furthermore, in mice models of early rheumatoid arthritis and 2,4,6-trinitrobenzene sulfuric acid (TNBS)-induced colitis, it was demonstrated that vitamins might inhibit the induction of IL-17 A [375,376]. When vitamins were administered, the number of taxa that positively linked with the production of SCFAs (such as *Akkermansia*, *Lactobacillus*, *Parvibacter*, *Staphylococcus*, *and Corynebacterium*) significantly increased. These include the genera *Parvibacter*, *Lactobacillus*, and *Akkermansia*, whose abundance and mucin levels are positively associated. Furthermore, the decrease in pathogenic *Escherichia/Shigella* abundance and the increase in *Lactobacillus*, *Parvibacter*, and *Akkermansia* abundance may be the primary reasons behind the therapeutic effects of vitamin A supplementation on colitis. By enhancing mucin, ZO-1, and anti-inflammatory factor expression, such a change in the gut microbiota may also lead to an increase in the gut barrier. Furthermore, an increase in intestinal SCFA levels may suppress the proliferation of pathogenic taxa in the gut microbiota and encourage the release of the anti-inflammatory cytokine IL-10 [377].

Certain species of *Lactobacillus*, *Bacteroides*, and *Bifidobacteria* possess the capability to synthesize water-soluble B vitamins such as riboflavin (vitamin B2), niacin (vitamin B3), and folate (vitamin B9) [378]. Unlike the vitamins obtained from the diet, which are primarily absorbed in the small intestine, microbial-derived vitamins are absorbed in the colon, where they can interact with mucosal immune cells [103]. Vitamin B2 metabolites play a critical role in modulating host defense mechanisms, as mucosal-associated invariant T (MAIT) cells recognize derivatives of vitamin B2 presented via the MHC-like protein MR1 on antigen-presenting cells in the colon [379]. In response to changes in bacterial growth, MAIT cells can directly adjust their activity by sensing variations in riboflavin levels, leading to increased expression of CD69, CD25, and PD-1 upon stimulation with *Escherichia coli* in the growth phase [380].

Similarly, niacin can signal to colonic CD103+ dendritic cells in mice through the engagement of the cell surface receptor GPR109a [381]. This signaling pathway promotes a regulatory T cell phenotype upon T cell engagement, further emphasizing the significant role of microbial compounds in maintaining immune homeostasis within the colon [382].

Through microbiome research, we can gain profound insights into how these ecosystems profoundly impact human health, both in fostering wellness and instigating disease processes. This underscores the microbiome’s potential as a therapeutic target for managing various conditions. There are additional diverse strategies aimed at modulating microbiome composition and functionality to achieve therapeutic objectives. As our understanding of the microbiome’s mechanistic roles in health deepens through ongoing research, the refinement of therapeutic interventions becomes increasingly feasible. Vitamins A (as well as the related family of carotenoid compounds), C (ascorbate), and E are examples of exogenous antioxidants that are not produced by our bodies [383]. The two subgroups of vitamin A are dehydroretinol (vitamin A2) and retinol (vitamin A1), and they maintain a structural relationship with the pro-vitamin A molecule β-carotene [384]. The antioxidant mechanisms of these compounds are not the same. The greater family of vitamins known as the vitamin A family may react with peroxyl radicals to act as a chain-breaking antioxidant before the radicals interact with lipids and produce hydroperoxides, which would otherwise cause cellular damage [385]. Singlet oxygen and peroxyl radicals are very reactive and unstable and can be scavenged by carotenoids [386]. Furthermore, by improving SOD and catalase activities, carotenoids could cause indirect antioxidant effects [387]. Research conducted both in vivo and in vitro has shown that beta-carotene protects against cancer [388]. Studies utilizing chemical carcinogens to induce malignant transformation in mouse mammary cell organ cultures revealed that beta-carotene supplementation significantly suppressed the incidence of neoplastic alterations [389]. Additionally, it has been demonstrated that the administration of beta-carotene prevents Chinese hamster ovary cells with chromosomal abnormalities from producing genotoxic compounds [389]. Similarly, retinyl acetate is useful against cutaneous and breast tumors in mice, although it is not very effective against these tumors in the rat model of breast cancer [390]. Apart from the numerous in vivo investigations that show vitamin A precursors have anticancer properties, a lot of studies also suggest that they may have an antioxidant mechanism. One form of ROS that can oxidize hemoglobin and damage phospholipids that are found in the red blood cell (RBC) membrane is superoxide, which can lead to a buildup of phospholipid hydroperoxides. Mice were given all-trans β-carotene (BC) at 6 g/kg or 30 g/kg for one week along with their usual semi-synthetic food in a study conducted by Nakagawa et al. [391]. Supplementing BC in the diet considerably reduced the formation of phospholipid hydroperoxides in mouse red blood cells when compared to the control group. But this potent antioxidant property action was seen only in the RBCs—not in the plasma, liver, or lungs [391]. This implies that the antioxidant effects of vitamin A generated from BC may target specific tissues or cells. Notably, all groups of mice consumed a typical semi-synthetic diet that included 50 mg of alpha-tocopherol/kg; also, there was no control group that examined the effects of an all-trans BC diet on its own. Consequently, it is hard to overlook any possible confusing effects of vitamin E on anti-phospholipid hydroperoxidation and vitamin A. 

Vitamin C, a popular exogenous supplement, has a well-established preventive role against tumorigenesis and possesses the capacity to scavenge free radicals [392]. Through its interaction with glutathione, vitamin C is kept in its reduced state, helping it to eliminate and diminish ROS [393,394]. Glutathione and reducing equivalents are two important ways that vitamin C may restore vitamin E in lipid membranes. 

Vitamin C is an essential cofactor for numerous classes of hydroxylases involved in controlling the transcription factor hypoxia-inducible factor 1 (HIF1) [395]. Because of the tumor cell’s capacity for rapid cell division and the promotion of poor blood vessel formation, elevated HIF activity may enhance the stem cell phenotype, which increases the lethality of the malignancy. To inhibit the growth of tumors and regulate HIF, HIF hydroxylases need to mark a protein for destruction. Because vitamin C serves as a cofactor for the HIF hydroxylases, HIF transcription activity increases and HIF hydroxylase activity decreases in cells lacking in vitamin C acid [395]. Generally, DNA showed less oxidative damage in cells treated with the antioxidant vitamin C than in the control group. According to some research, vitamin C acts as an antioxidant to stop oxidative damage, which may lessen the risk of tumor development. Applying cisplatin treatment, Leekha et al. examined the anticancer effects of vitamin C on cervical cell lines SiHa and control cell lines HEK293 [396]. Researchers examined the cytotoxicity of vitamin C and cisplatin at different concentrations both alone and in synergy in cervical cancer cells. For cisplatin, the dosage ranged from 5 to 200 mM, while for vitamin C, it was 25, 50, and 100 μg/mL for 24, 48, and 72 h. The cytotoxicity assay utilized the following combinations for time periods of 24, 48, and 72 h: 100 mM cisplatin + 100 μg/mL vitamin C; 50 mM cisplatin + 100 μg/mL vitamin C; 5 mM cisplatin + 100 μg/mL vitamin C; 1 mM cisplatin + 100 μg/mL vitamin C; and 50 mM cisplatin + 50 μg/mL vitamin C [396]. In their combined treatment of the cervical cancer cell line SiHa, the antioxidant vitamin C and cisplatin indicated synergistic amplification in cell death. This proves that vitamin C stimulates the death of tumor cells while becoming specific to cancer cells [397].

Groups of related tocopherols and tocopherols are all referred to simply as vitamin E [398]. According to its higher bioavailability, α-tocopherol has been the topic of several studies [399]. The lipid peroxidation chain reaction produces lipid radicals, and this lipid-soluble oxidant acts to avoid lipid membrane oxidation [400]. The neutralization of free radicals by this process inhibits the peroxidation reaction from escalating and harming cell membranes. As a result of this reaction, oxidized α-tocopherol is produced, which interacts with ascorbates to be reduced and utilized back into its antioxidant state [401]. Retinol may also combine with free radicals called tocopheryl radicals and produce new α-tocopherol [385]. 

In lipopolysaccharide (LPS)-activated macrophages and many tumor cell lines, γ-tocotrienol was an effective inhibitor of NF-κB activation. By suppressing NF-κB activity with γ-tocotrienol, inflammation decreases in these LPS-stimulated macrophages by diminishing the generation of IL-6 and granulocyte-colony stimulating factor (G-CSF) [402,403]. γ-tocotrienol has also been demonstrated to activate protein-tyrosine phosphatase SHP-1, suppressing the JAK-STAT3 signaling pathway in cancer cells [404]. Finally, by preventing STAT6 from being phosphorylated and binding to DNA, γ-tocotrienol can inhibit JAK-STAT6 signaling [402]. Because of these considerations, vitamin E changes have become attractive as prospective adjuncts to several cancer treatments and prophylactic measures. 

On the other hand, recent evidence demonstrates that high vitamin E consumption may raise the risk of certain cancers, including prostate cancer, by influencing the expression of different cytochrome P450 enzymes [405]. Both 0.17% and 0.3% α-tocopherol supplementation proved to be helpful in terms of growth inhibition, tumor volume, and tumor weight, and there was no discernible decrease from the non-supplemented control. In contrast to the non-supplemented control, 0.3% α-tocopherol supplementation seldom caused an effect on tumor volume and weight reduction [406]. Vitamin E forms reduced LPS-stimulated IL-6 in macrophages, as demonstrated by several investigations [403,407]. However, IL-10, an anti-inflammatory cytokine, was not significantly affected by vitamin E forms [407]. By reducing the activation of NF-κB caused by LPS and upregulating C/EBPβ and C/EBPδ, mechanistic investigations have shown that γTE inhibits IL-6 [407]. In cancer cell lines, γTE also prevents NF-κB activation [408]. In human lung epithelial A549 cells, γTE is more efficient than other forms of vitamin E in blocking the production of eotaxin-3 induced by interleukin-13 (IL-13) by limiting the phosphorylation of STAT6 along with binding to DNA [409]. Asthma’s hallmark, lung eosinophilia, is induced by eotaxins-3 (CCL26), a crucial chemokine in the pathophysiology of asthma. Mechanistic analysis showed that γ-TE promoted PAR4 (prostate-apoptosis-response 4), which in turn inhibits atypical protein kinase C (aPKC)-mediated STAT6 activation, hence inhibiting IL-13/STAT6-activated eotaxin. Also, it has been demonstrated that γTE inhibits JAK1-STAT3 signaling on multiple types of cancer cells by stimulating protein-tyrosine phosphatase SHP-1 [406]. Zingg et al. [410] investigated the possible impact of αT and γT on CD3/CD28-stimulated gene expression in spleen T cells isolated from elderly mice fed with these tocopherols at a low dose (30 mg/kg) or high dose (500 mg/kg). Supplementing with γT resulted in the inhibition of cytokines, chemokines, and signaling lymphocytic activation molecules generated by CD3/CD28, as opposed to high-dose αT. Because of this, γT seems to inhibit gene upregulation following T cell activation more potently than αT [410]. Along with colon cancer, the γT-enriched diet was found to reduce ventral prostate epithelial dysplasia and attenuate COX-2 and matrix metalloproteinase (MMP-9) activity upregulation produced by N-methyl-N-nitrosourea (MNU) in rats [411]. Barve et al. [412] showed that in the mouse prostate cancer TRAMP model, γT-rich mixed tocopherols preserved redox-sensitive transcription factor Nrf2 and Nrf2-regulated antioxidant genes while suppressing the incidence of palpable tumors.

In conclusion, different investigations have shown that antioxidant supplements containing vitamins lower the risk of specific cancer types. In contrast, some evidence suggests that taking antioxidant supplements can increase the risk of developing cancer.

## 14. Conclusions

We can conclude that the intricate relationship between the gastrointestinal (GI) microbiome and the progression of chronic non-communicable diseases underscores the significance of developing strategies to modulate the GI microbiota for promoting human health. The key findings of this review are as follows:The human microbiome is found in niches that mimic the natural environment of the body [31]. Environmental disturbances can cause changes in the variety and content of microbes, altering the balance of the microbiome and possibly exposing people to certain disease states [30]. The importance of creating methods to change the GI microbiota in order to promote human health is highlighted by the complex interaction that exists between the development of chronic non-communicable illnesses and the GI microbiome. In this sense, probiotic supplements are a promising method for changing the structure of the microbiome to treat various illnesses; however, more studies are needed to clarify the mechanisms and optimize treatment results.Probiotics and prebiotics aim to enhance the population of beneficial bacteria in the intestinal lumen post-consumption [118]. Probiotics introduce helpful bacteria into our bodies, whereas prebiotics provide fermentable carbohydrates that specifically encourage the growth of good bacteria already present in the gut [116]. Prebiotic polysaccharides are dietary fibers that are digested by the large intestinal microbiota. This process improves the health of the intestinal mucosa, increases fecal weight and biomass, controls the frequency of defecation, and maintains overall gut health [122].The identification of metabolites related to diseases may offer simpler insights, which may help in the application of synthetic methods for therapeutic interventions [128]. Interestingly, the *Prevotella/Bacteroides* ratio changes throughout societies with different dietary practices, suggesting the impact of long-term dietary variations, such as diets high in meat in Westernized populations and high in fiber in non-Westernized populations [138].Dietary fibers are an important source of energy for the bacteria that live in the colon and cecum. Anaerobic bacteria can digest complex carbohydrates by activating the enzyme machinery and metabolic pathways in certain intestinal circumstances. This process produces metabolites like SCFAs [67]. Intestinal mucosal health depends on SCFAs, mainly acetate, propionate, and butyrate, which are not easily obtained from food.Dietary interventions high in dietary fiber could be very important in preventing cancer, particularly CRC. A plethora of research highlights the critical function that dietary fiber intake plays in maintaining general metabolic health, operating via basic pathways such as GPCRs, Wnt signaling, and T regulatory (Treg) cells. Here, we present unambiguous links between dietary fiber consumption and enhanced gut motility, improved colon function, and reduced risk of CRC. It has become clear that the gut microbiota plays a crucial mediating role in the health benefits of dietary fiber, impacting areas including chronic inflammation pathways and metabolic functions. The normal Western diet is low in dietary fiber for a number of reasons. The majority of people have adapted to modern lives that are defined by diets high in ultra-processed foods. But unlike our bodies, our gut flora has not adapted to this shift in nutrition. The benefits of our food choices are mediated by our gut bacteria, and they are a predictor of our general health and well-being.Prominent concerns encompass the excessive consumption of fats and sugars, along with the conspicuous absence of dietary fiber in contemporary diets. Future health results stand to benefit greatly from consumers choosing high-fiber diets versus ultra-processed, low-fiber ones. This change in consumer tastes will likely have an impact on food companies’ strategic commercial objectives, leading to an increase in processed foods’ fiber content. The state of total health is reflected in the reciprocal and bidirectional relationships among food, dietary antioxidants, inflammation, and body composition (obesity).Only a few bacterial species, such as *Bifidobacteria* and *Lactobacillus*, which are regarded to be beneficial for human gut health, were found to be impacted by dietary polyphenol intake. Numerous polyphenols have been shown to have an inhibitory impact on *Clostridium* and *Bacteroides*, two types of harmful bacteria [282].The most common mechanism for the action of antioxidants is as follows: (a) chelating trace elements or inhibiting the enzymes that generate free radicals to decrease the formation of ROS; (b) scavenging ROS; and (c) upregulating antioxidant enzymes that offer protection [317]. Because of their antioxidant activity, flavonoids can safeguard living things. Certain forms of cancer can be prevented by taking vitamin-containing antioxidant supplements, according to various studies. Taking antioxidant supplements, however, may raise the risk of cancer, according to some research.

## Figures and Tables

**Figure 1 ijms-25-08250-f001:**
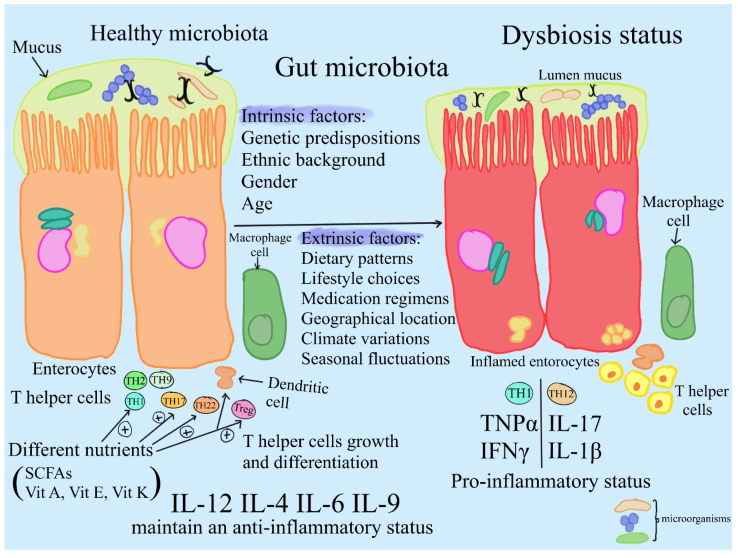
A comparison of dysbiosis and the healthy microbiome in humans. In particular, several factors (intrinsic and extrinsic), including age, nutrition, environmental conditions, and antibiotic use, significantly influence how the intestinal microbial community of humans is modulated.

**Figure 2 ijms-25-08250-f002:**
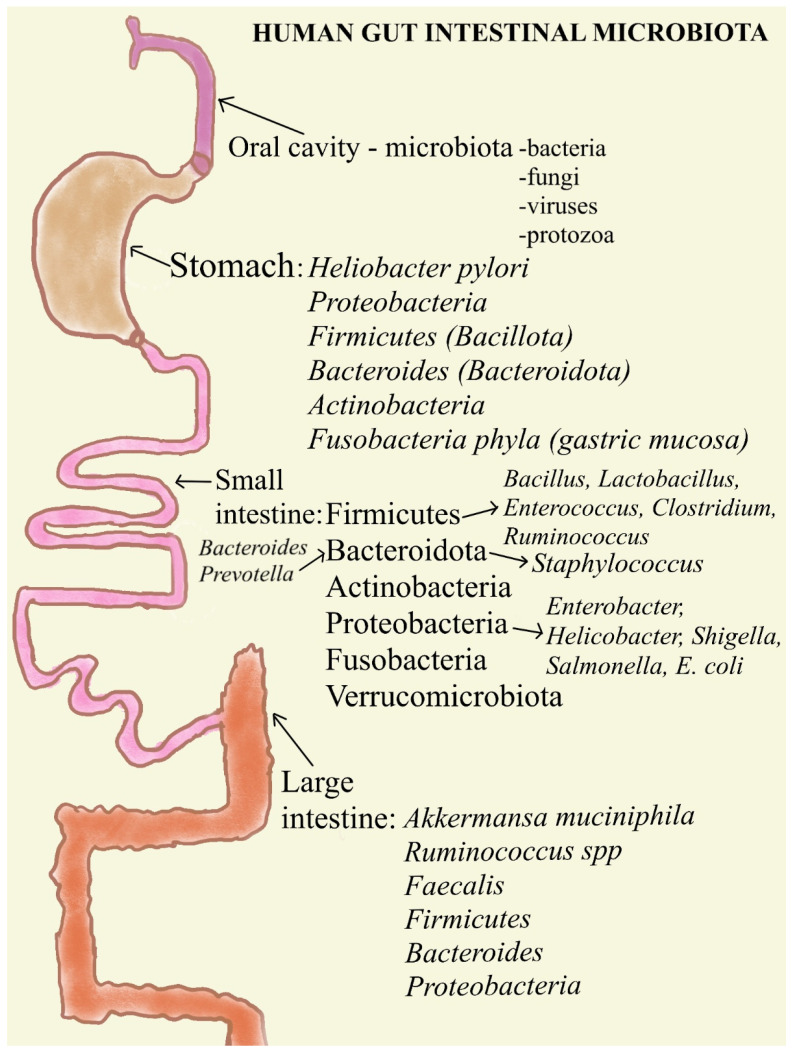
The oral microbiome, which acts as a major entry point into the body, is important for both health and illness, impacting physiological functions and systemic immunity. Notwithstanding difficulties with diagnostic techniques, discoveries have revealed a varied gastric microbiome that is primarily made up of the phyla Proteobacteria, Firmicutes, Bacteroidetes, Actinobacteria, and Fusobacteria in the gastric mucosa. Gastric juice contains a unique microbial community that differs from mucosal populations. It is mostly composed of Firmicutes, Actinobacteria, and Bacteroidetes, with Proteobacteria and Firmicutes predominating in mucosal habitats. The transitory colonization of the stomach microbiota by oral and duodenal bacteria such as *Lactobacillus*, *Clostridium*, and *Veillonella* emphasizes this dynamic aspect of the microbiome. The stomach environment is significantly altered by *Helicobacter pylori*, which is common in persons with infections and may affect resident microbial ecosystems. Commensal bacteria are mostly found in the colon, with smaller populations found in the stomach and small intestine. The gut microbiota is primarily composed of the phyla Firmicutes and Bacteroides, although Actinobacteria, Proteobacteria, Fusobacteria, and Verrucomicrobia are also present. With its pH ranging from neutral to slightly acidic and its slow flow rates, the large intestine is home to the biggest microbial community, which is mainly made up of obligate anaerobes.

**Figure 3 ijms-25-08250-f003:**
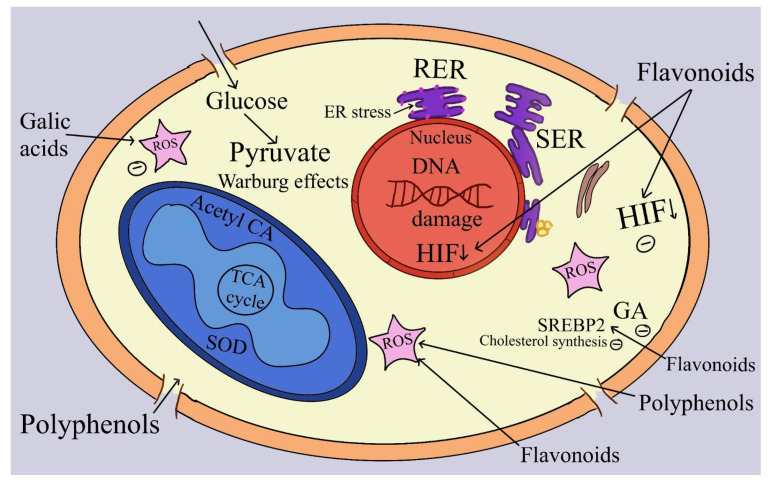
The influence of antioxidant compounds on tumor cells. Specifically, tumor cells show an anabolic growth program that produces lipid, protein, and nucleotide synthesis, allowing the Warburg effect to be manifested. By transcriptionally regulating metabolic genes, the loss of tumor suppressors such as p53 further enhances anabolism. Metabolism manages acetylation, methylation, and reactive oxygen species (ROS) production in order to modulate signaling. In conclusion, the effects of the different antioxidant compounds are evidenced to be beneficial to cells.

**Table 2 ijms-25-08250-t002:** Modulating microbiome composition through prebiotic supplementation.

Prebiotics	Effects	References
	1. Prebiotics can be used to transform the microbiome from a sick state to a healthy one by encouraging the growth of helpful microorganisms.	[117]
	2. The human gut has shown that XOSs can increase the growth of *Bifidobacterium*; however, because intervention doses and durations vary, the effects on other bacterial genera are still unknown.	[118]
	3. Preclinical research, notably, has shown that the gut microbiome of individuals who respond to immune checkpoint blockers contains more beneficial bacteria than that of non-responders, including *Bifidobacterium*, *Akkermansia*, *Ruminococcaceae*, and *Faecalibacterium*.	[120]
	4. Beneficial bacteria including *Lactobacillus*, *Roseburia*, and *Akkermansia* increased in mice given oral inulin gel, according to research by Han et al. Combined with α-PD-1, this boosted the antitumor effects by inducing a T cell response.	[121]

XOS, xylo-oligosaccharides; α-PD-1, programmed cell death protein 1.

## Data Availability

The original contributions presented in the study are included in this article. Further inquiries can be directed to the corresponding authors.

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
