# Peer review of "Interactions between Dietary Antioxidants, Dietary Fiber and the Gut Microbiome: Their Putative Role in Inflammation and Cancer"

_ijms, 2024, doi:10.3390/ijms25158250_

Round 1

Reviewer 1 Report

Comments and Suggestions for Authors

Dear Sirs,

The authors of this manuscript explore the interactions between dietary antioxidants, dietary fiber, and the gut microbiome, with a particular emphasis on their potential roles in modulating inflammation and influencing malignancy. Their review is comprehensive and well-written in parts. However, I recommend the following improvements:

Lines 19-23: Please rephrase as follows:

"They are known for their antioxidant properties and their ability to inhibit angiogenesis, metastasis, and cell proliferation. Additionally, they promote cell survival, modulate immune and inflammatory responses, and inactivate pro-carcinogens. These actions collectively contribute to their role in cancer prevention."

Line 28: Please add a sentence regarding the aim of this review.

Line 72 (Figure 1): Please correct the following:

- it should be "dendritic cell"

- it should be "IL-12" etc

There should be a gap between "Th17" and "Th22"

Line 151: Please rephrase as follows:

"Helicobacter pylori, prevalent in infected individuals, orchestrates significant alterations in the gastric environment, potentially impacting resident microbial habitats."

Line 247: It should be "Clostridioides difficile."

Line 284: There is conflicting evidence regarding the role of the F/B ratio as a biomarker for obesity, as recent data have shown no association between the F/B ratio and obesity.

Line 274: Under the heading "Changing the Population Dynamics of the Microbiome," data regarding the role of next-generation probiotics and prebiotics should be added.

Additional Queries: Is there any connection between diet, the gut microbiome, and epigenetics? If so, this should be addressed.

Additional comments:

-The authors should add a "Literature Search Methodology" section immediately after the introduction.

- The illustrations of the figures should be improved.

- The authors should ensure consistency in abbreviations throughout the manuscript. For example, in line 538, "short chain fatty acids" should be removed.

Comments on the Quality of English Language

Minor editing of English language required. 

Author Response

R1

We thank you for your suggestions and comments, dear reviewer. We give our response to the queries in red, and the reviewers’ are in black.

Dear Sirs,

The authors of this manuscript explore the interactions between dietary antioxidants, dietary fiber, and the gut microbiome, with a particular emphasis on their potential roles in modulating inflammation and influencing malignancy. Their review is comprehensive and well-written in parts. However, I recommend the following improvements:

Lines 19-23: Please rephrase as follows:

"They are known for their antioxidant properties and their ability to inhibit angiogenesis, metastasis, and cell proliferation. Additionally, they promote cell survival, modulate immune and inflammatory responses, and inactivate pro-carcinogens. These actions collectively contribute to their role in cancer prevention."

Thanks for your suggestion. As you asked us, we moved this sentence to the abstract.

Line 28: Please add a sentence regarding the aim of this review.

Thank you very much. You are right. We have put the purpose at both the end of the abstract and the end of the introduction.

Line 72 (Figure 1): Please correct the following:

- it should be "dendritic cell"

- it should be "IL-12" etc

There should be a gap between "Th17" and "Th22"

As you suggested, we made the corrections. Thank you again.

Line 151: Please rephrase as follows:

"Helicobacter pylori, prevalent in infected individuals, orchestrates significant alterations in the gastric environment, potentially impacting resident microbial habitats."

We rephrased as you suggested.

Line 247: It should be "Clostridioides difficile."

Thank you. We wrote as you corrected.

Line 284: There is conflicting evidence regarding the role of the F/B ratio as a biomarker for obesity, as recent data have shown no association between the F/B ratio and obesity.

Excellent suggestion. We reworded the paragraph as a debate. We are sure that now our manuscript has improved.

Line 274: Under the heading "Changing the Population Dynamics of the Microbiome," data regarding the role of next-generation probiotics and prebiotics should be added.

As you suggested, we have added data on the role of next-generation probiotics and prebiotics. Thanks a lot.

Additional Queries: Is there any connection between diet, the gut microbiome, and epigenetics? If so, this should be addressed.

Excellent appreciation. We addressed the connection between diet, gut microbiome, and epigenetics at the end of chapter 4: Diet and Microbiome.

Additional comments:

-The authors should add a "Literature Search Methodology" section immediately after the introduction.

As you suggested, we put the main text in the "Literature Search Methodology" section immediately after the introduction. Thank you!

- The illustrations of the figures should be improved.

We improved the figures. Thank you for the suggestion.

- The authors should ensure consistency in abbreviations throughout the manuscript. For example, in line 538, "short chain fatty acids" should be removed.

You are right. As you recommended, we ensured the consistency of the abbreviations. Thanks a lot!

Reviewer 2 Report

Comments and Suggestions for Authors

There is a lot of experimental data and indirect proofs on links between microbiome composition, antioxidants and cancer in relation to different organs. They are summarized on a dozen of systematic and narrative reviews. This ms belongs to the latter category and its scope is limited to gut microbiome and influences on its composition exerted by antioxidants consumed in a human diet.

The paper is undoubtedly extraordinary in the sense of its length: 2361 lines, 48 pages which makes any review rather superficial. The text is somehow no consistent; important clues are contained in separated sentences supported with single citations only. It seems that the authors addressed their review to readers not adequately informed about microbiome, antioxidants, colon cancer etc. It is also possible that the text originates from a thesis or monography. The literature cited is not fully actual; the latest citations are dated for 2022. What is even more important is that the link between dietary antioxidants and gut microbiome is not properly demonstrated and explained although they are no longer putative. Data on human microbiome presented in this review are not fully actual it seems that they have been collected in the pre-NGS era.

There are several chapters in this review which are unrelated to the general line. For example:  

- The paragraph on CDI (Clostridioides difficile infection); moreover the actual view on this infection is different than presented in the ms since actual data on mechanisms and epidemiology of CDI show that both are much more complicated.

-              Same is true for CS (Cesarean Section) and acquisition of the neonatal microbiome and infections related to neonatal gut microbiota alterations: is there any proven relation to gut cancer? Or to dietary fiber?    

Last remark: Species of bacteria in Latin should be in italics.

Author Response

R2

We thank you for your suggestions and comments, dear reviewer. We give our response to the queries in red, and the reviewers’ are in black.

There is a lot of experimental data and indirect proofs on links between microbiome composition, antioxidants and cancer in relation to different organs. They are summarized on a dozen of systematic and narrative reviews. This ms belongs to the latter category and its scope is limited to gut microbiome and influences on its composition exerted by antioxidants consumed in a human diet.

The present study centers on the Interactions between Dietary Antioxidants Dietary Fiber and Gut Microbiome related to inflammatory diseases and cancer. Food antioxidants are essential nutrients crucial for maintaining human health by counteracting oxidative stress resulting from diverse biochemical and metabolic processes. The interplay between gut microbiota and antioxidants is pivotal for sustaining gut microbiome homeostasis, promoting microbial diversity, and enhancing the bioavailability of antioxidants through the production of bioactive metabolites by the microbiota. We mentioned herein numerous studies that underscore the dynamic interaction between food-derived antioxidants and the gut microbial population, highlighting their role in mitigating gut microbial dysbiosis.

We show that clinical investigations consistently demonstrate the beneficial impact of dietary antioxidants on gut microflora, suggesting their potential for developing microbiome-targeted therapies for various ailments. This review comprehensively examines the health benefits associated with major groups of dietary antioxidants and delves into the intricate interactions between gut microbiota and these antioxidants, incorporating insights from preclinical models and human dietary interventions. A similar approach was conducted towards the Interactions between Dietary Fiber and Gut Microbiome related to inflammatory diseases and cancer.

The paper is undoubtedly extraordinary in the sense of its length: 2361 lines, 48 pages which makes any review rather superficial. The text is somehow no consistent; important clues are contained in separated sentences supported with single citations only. It seems that the authors addressed their review to readers not adequately informed about microbiome, antioxidants, colon cancer etc. It is also possible that the text originates from a thesis or monography.

We accept that this review is very extensive since the themes discussed are broad and complicated and various interplays are detected between all the subjects examined. We think that the text flows swiftly and updated citations are introduced. Thus, this review is targeted to readers well informed as well as not adequately informed about interactions of microbiome, antioxidants, dietary fiber, inflammation and colon cancer. We inform that the present text was not originated nor in a thesis nor in a monography.

The literature cited is not fully actual; the latest citations are dated for 2022.

We have added some citations dated 2023 and 2024.

As you suggested we modified the main text to clarify that the references are up to date. We added the "Literature Search Methodology" section. Probably through this section, makes it easier to see that our bibliographic sources are recent. More than that, we have added at the end of chapter 2 presented data on the role of next-generation probiotics and prebiotics. Thanks a lot.

What is even more important is that the link between dietary antioxidants and gut microbiome is not properly demonstrated and explained although they are no longer putative. Data on human microbiome presented in this review are not fully actual it seems that they have been collected in the pre-NGS era.

We present herein pivotal findings that offer insights for guiding future investigations using Next-Generation Sequencing (NGS) of the microbiome and implementation research. We present a meticulous examination of individual diet-health relationships as presented by Zeevi et al. who identified significant variability in human dietary metabolic processing and physiological responses (Zeevi D, Korem T, Zmora N, Israeli D, Rothschild D, Weinberger A, Ben-Yacov O, Lador D, Avnit-Sagi T, Lotan-Pompan M, Suez J, Mahdi JA, Matot E, Malka G, Kosower N, Rein M, Zilberman-Schapira G, Dohnalová L, Pevsner-Fischer M, Bikovsky R, Halpern Z, Elinav E, Segal E. Personalized Nutrition by Prediction of Glycemic Responses. Cell. 2015 Nov 19;163(5):1079-1094.).

The link between probiotics metabolome is now addressed on a recent study demonstrating that Lactobacillus plantarum-derived indole-3-lactic acid ameliorates colorectal tumorigenesis via epigenetic regulation of CD8+ T cell immunity. This is a recent study (the citation is al (Zhang Q, Zhao Q, Li T, Lu L, Wang F, Zhang H, Liu Z, Ma H, Zhu Q, Wang J, Zhang X, Pei Y, Liu Q, Xu Y, Qie J, Luan X, Hu Z, Liu X. Lactobacillus plantarum-derived in-dole-3-lactic acid ameliorates colorectal tumorigenesis via epigenetic regulation of CD8+ T cell immunity. Cell Metab. 2023 Jun 6;35(6):943-960.).

Additionally, we cite a recent study directed towards the relationship between GI ROS, dietary antioxidants, and the beneficial effect on the survival of a specific beneficial gut bacteria (Van Buiten CB, Seitz VA, Metcalf JL, Raskin I. Dietary Polyphenols Support Akkermansia muciniphila Growth via Mediation of the Gastrointestinal Redox Environment. Antioxidants (Basel). 2024 Feb 29;13(3):304).

There are several chapters in this review which are unrelated to the general line. For example: 

- The paragraph on CDI (Clostridioides difficile infection); moreover, the actual view on this infection is different than presented in the ms since actual data on mechanisms and epidemiology of CDI show that both are much more complicated.

We have modified the descriptions on the complex mechanisms of CDI. Thank you. 

Same is true for CS (Cesarean Section) and acquisition of the neonatal microbiome and infections related to neonatal gut microbiota alterations: is there any proven relation to gut cancer? Or to dietary fiber?    

As you asked us, we answered these questions.

Last remark: Species of bacteria in Latin should be in italics.

All species bacteria are now on italics.

Reviewer 3 Report

Comments and Suggestions for Authors

This is an interesting review article with adequate novelty. Some points should be addressed.

- Concerning the Abstract, it could be better to include distinct subheadings (e.g. Background, Methods, Results, Conclusions.

- In the Abstract, the authors should provide a brief description of the methodology used to collect the relevant data for their review.

- Accordingly, the authors should emphasize tha aim of their review article before reporting the methodology used to collect their data.

- The paragraph in lines 75-80 of the Introduction section focuses on an important topic. So, the authors could provide a bit more information about this topic, adding some additional relevant references.

- At the end of the introduction section, the authors should provide the literature gap which their review article will cover. Afterwards, the authors should report the main aims of their review article.

- The paragraph in lines 157-159 needs a bit more analysis as it presents a very interesting topic.

- The paragraph in lines 216-220 also needs a bit more analysis as it presents a very interesting topic.

- Accordingly, the paragraphs in lines 236-239 and 240-244 need a bit more analysis.

- The paragraph in lines 585-589 also needs a bit more analysis, especially concerning the butyrate for which thera are several information concerning its beneficial effects in human health promotion.

- The paragraph in lines 751-813 is too long and should be split into two smaller and readable paragraphs.

- The paragraph in lines 818-843 should also be split into two smaller paragaphs.

- At least 1-2 references are needed for the last 3 sentences in lines 840-844.

- Please check if all the abbreviations used are explained.

-  The sentences in lines 1180-1218 is quite long  and chould be split into two smaller paragraphs.

- In the sentences of lines 1089-1093, relevant references should be added.

- The paragraph in lines 1180-1218 is a bit long and it should be split into to smaller paragraphs.

Comments on the Quality of English Language

Moderate editing of English language is required.

Author Response

R 3

We thank you for your suggestions and comments, dear reviewer. We give our response to the queries in red, and the reviewers’ are in black.

This is an interesting review article with adequate novelty. Some points should be addressed.

- Concerning the Abstract, it could be better to include distinct subheadings (e.g. Background, Methods, Results, Conclusions.

As you suggested, we added information to distinguish these distinct subheadings. Thank you!

- In the Abstract, the authors should provide a brief description of the methodology used to collect the relevant data for their review.

Excellent suggestion. As you recommended we added the section Literature Search Methodology.

- Accordingly, the authors should emphasize the aim of their review article before reporting the methodology used to collect their data.

You are right. We added before section Literature Search Methodology the aim of the manuscript.

- The paragraph in lines 75-80 of the Introduction section focuses on an important topic. So, the authors could provide a bit more information about this topic, adding some additional relevant references.

Thank you! As you suggested, we added more information about this topic.

- At the end of the introduction section, the authors should provide the literature gap which their review article will cover. Afterwards, the authors should report the main aims of their review article.

We made all these recommendations. We are sure that our paper was improved after that.

- The paragraph in lines 157-159 needs a bit more analysis as it presents a very interesting topic.

Thanks a lot for the suggestion. Now this paragraph is analyzed more in depth.

- The paragraph in lines 216-220 also needs a bit more analysis as it presents a very interesting topic.

As you recommended, we have improved this paragraph. Thank you!

- Accordingly, the paragraphs in lines 236-239 and 240-244 need a bit more analysis.

You are right. As you recommended, we added more information about the subject. Thank you.

- The paragraph in lines 585-589 also needs a bit more analysis, especially concerning the butyrate for which thera are several information concerning its beneficial effects in human health promotion.

You are right. As you recommended, we added more information about butyrate and its beneficial effects on human health promotion.

- The paragraph in lines 751-813 is too long and should be split into two smaller and readable paragraphs.

You are right. As per your suggestion, we have divided this paragraph into several shorter ones. Thank you!

- The paragraph in lines 818-843 should also be split into two smaller paragraphs.

We also split this paragraph into several shorter ones. Thank you!

- At least 1-2 references are needed for the last 3 sentences in lines 840-844.

As you recommended, we added 3 references in this paragraph.

- Please check if all the abbreviations used are explained.

We explained all the abbreviations. Thank you!

-  The sentences in lines 1180-1218 is quite long  and chould be split into two smaller paragraphs.

You are right. As per your suggestion, we have divided this paragraph into several shorter ones. Thank you!

- In the sentences of lines 1089-1093, relevant references should be added.

You are right. We added the relevant references at the end of these sentences.

- The paragraph in lines 1180-1218 is a bit long and it should be split into to smaller paragraphs.

You are right. As per your suggestion, we have divided this paragraph into several shorter ones. Thank you!

Round 2

Reviewer 1 Report

Comments and Suggestions for Authors

The recommended changes have been made. The manuscript has been improved and may be published.

Comments on the Quality of English Language

The quality of English language is acceptable.